# Skill Reinforcement Learning and Planning for Open-World Long-Horizon Tasks

## Abstract

Developing autonomous agents with multi-task capabilities in open-world environments has been a longstanding goal of AI research. Without human demonstrations, learning to accomplish long-horizon tasks in a large open-world environment with reinforcement learning (RL) is extremely inefficient. To tackle this challenge, we convert the multi-task learning problem into learning basic skills and planning over the skills. We employ RL with intrinsic rewards, enabling the agent to acquire a set of basic skills. These skills can be reused and chained together to solve diverse long-horizon tasks. Given the challenge of exploring large open-world environments using RL, we propose a novel Finding-skill that aims at finding target items of subsequent skills and providing effective state initialization for these skills. In skill planning, we utilize the prior knowledge in Large Language Models (LLMs) to construct a skill graph that depicts the relationships between skills. When solving a task, at each stage, the agent searches for a path on the skill graph and executes the first skill. In the popular open-world game Minecraft, our method accomplishes 40 diverse tasks, where many tasks require sequentially executing more than 10 skills. Our method outperforms baselines by a large margin and is the most sample-efficient demonstration-free RL method to solve Minecraft Tech Tree tasks.

## 1 Introduction

Learning diverse tasks in open-ended worlds is a significant milestone toward building generally capable agents. Recent studies in multi-task reinforcement learning (RL) have achieved great success in many narrow domains like games (Schrittwieser et al., 2020) and robotics (Yu et al., 2020). However, transferring prior methods to open-world domains (Team et al., 2021; Fan et al., 2022) remains unexplored. Minecraft, a popular open-world game with an infinitely large world size and a huge variety of tasks, has been regarded as a challenging benchmark (Guss et al., 2019; Fan et al., 2022).

Previous works usually build policies in Minecraft upon imitation learning, which requires expert demonstrations (Guss et al., 2019; Cai et al., 2023; Wang et al., 2023c) or large-scale video datasets (Baker et al., 2022; Lifshitz et al., 2023; Yuan et al., 2024). Without demonstrations, RL in Minecraft is extremely sample-inefficient. A state-of-the-art model-based method (Hafner et al., 2023) takes over 10M environment steps to harvest cobblestones 🪨, even if the block breaking speed of the game simulator is set to very fast additionally. This difficulty comes from at least two aspects. First, the world size is too large and the requisite resources are distributed far away from the agent. With partially observed visual input, the agent cannot identify its state or do effective exploration easily. Second, an open-world task usually has a long horizon, with many sub-goals. For example, mining a cobblestone involves more than 10 sub-goals (from harvesting logs 🪵 to crafting wooden pickaxes 🔨) and requires thousands of environment steps.

To mitigate the issue of learning long-horizon tasks, we propose to solve diverse tasks in a hierarchical fashion. We propose to learn a set of basic skills, each representing a simpler task with a short horizon. Then, solving a task can be decomposed into planning for a proper sequence of basic skills and executing the skills interactively. We train RL agents to acquire skills and build a high-level planner upon the skills.

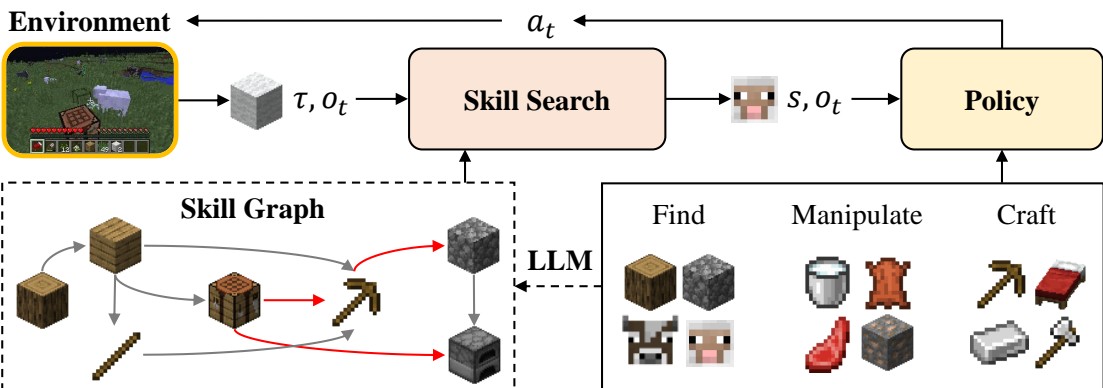

Figure 1: Overview of **Plan4MC**. We categorize the basic skills in Minecraft into three types: Finding-skill, Manipulation-skill, and Crafting-skill. We train policies to acquire skills with reinforcement learning. With the help of LLM, we extract relationships between skills and construct a skill graph, as shown in the dashed box. During online planning, the skill search algorithm walks on the pre-generated graph, decomposes the task into an executable skill sequence, and interactively selects policies to solve complex tasks.

We find that acquiring skills with RL still remains challenging due to the difficulty in finding the required resources in the vast world. As an example, if we use RL to train the skill of harvesting logs in Minecraft, the agent can always receive 0 reward through random exploration since it cannot find a tree nearby. On the contrary, if a tree is always initialized close to the agent, the skill can be learned efficiently (Table 1). Thus, we propose to pre-train a special Finding-skill that traverses the open world to find different items. This skill can provide better initialization for all other skills, improving the sample efficiency of learning other skills with RL. In Minecraft, our Finding-skill is implemented with a hierarchical policy, maximizing the area traversed by the agent.

In Minecraft, after integrating the Finding-skill, we classify all basic skills into three types: the Finding-skill, Manipulation-skills, and Crafting-skills. Each basic skill solves an atomic task that may not be further divided. Such tasks have a shorter horizon and require exploration in smaller regions of the world. Thus, using RL to learn these basic skills is more feasible. To improve the sample efficiency of RL, we introduce intrinsic rewards to train policies for different types of skills.

For high-level skill planning, recent works (Brohan et al., 2023; Wang et al., 2023c;a) demonstrate promising results via interacting with Large Language Models (LLMs). Though LLMs generalize to open-ended environments well and produce reasonable skill sequences, the unpredictable mistakes – given a specific prompt, it's challenging to foresee the errors an LLM planner might generate – becomes a challenging issue. Fixing these mistakes requires careful prompt engineering (Huang et al., 2022b; Wang et al., 2023c), involving iterative cycles of running the agent in the environment, identifying errors, and meticulously adjusting the prompt to mitigate these errors. This process is notably time-consuming. To make more flawless skill plans without much human engineering, we propose a complementary skill search approach. In the preprocessing stage, we use an LLM to generate the relationships between skills and construct a skill dependency graph. Then, given any task and the agent's condition (e.g., available resources/tools), we propose a search algorithm to interactively plan for the skill sequence. Figure 1 illustrates our proposed framework, **Plan4MC**.

In experiments, we use the MineDojo (Fan et al., 2022) simulator to set up 40 diverse tasks in Minecraft. These tasks involve executing diverse skills, including collecting basic materials 🪵🪨, crafting useful items 🪵🪨⛏️, and interacting with mobs 🐺🐷. Each task requires planning and execution for 2~30 basic skills and takes thousands of environment steps. Results show that Plan4MC accomplishes all the tasks and outperforms the baselines significantly. Also, Plan4MC can craft iron pickaxes ⛏️ in the Minecraft Tech Tree and is much more sample-efficient than existing demonstration-free RL methods.

To summarize, our main contributions are:

- To enable RL methods to efficiently solve diverse open-world tasks, we propose to learn fine-grained basic skills including a Finding-skill and train RL policies with intrinsic rewards. Thus, solving long-horizon tasks is transformed into planning over basic skills.

- Unlike previous LLM-based planning methods, we propose the skill graph and the skill search algorithm for interactive planning. The LLM only assists in the generation of the skill graph before task execution, avoiding unpredictable failures caused by the LLM.

- Our hierarchical agent achieves promising performance in diverse and long-horizon Minecraft tasks, demonstrating the great potential of using RL to build multi-task agents in open-ended worlds.

## 2 Preliminaries

### 2.1 Problem Formulation

In Minecraft, a task $\tau = (g, I)$ is defined with the combination of a goal $g$ and the agent's initial condition $I$, where $g$ represents the target entity to acquire in the task and $I$ represents the initial tools and conditions provided for the agent. For example, a task can be 'harvest cooked_beef 🍖 with sword 🗡 in plains'. We model the task as a partially observable Markov decision process (POMDP) (Kaelbling et al., 1998). The agent's observations include egocentric RGB images, as well as additional information such as lidar, location, biome, and life status. The actions consist of low-level mouse and keyboard controls, and the crafting process can be executed with a single action using the provided primitives. The control operates at a frame rate of about 30fps. At each timestep $t$, the agent obtains the partial observation $o_t$, takes an action $a_t$ following its policy $\pi(a_t|o_{0:t}, \tau)$, and receives a sparse reward $r_t$ indicating task completion. The agent aims to maximize its expected return $R = \mathbb{E}_\pi \sum_t \gamma^t r_t$.

To solve complex tasks, humans acquire and reuse skills in the world, rather than learn each task independently from scratch. Similarly, to solve the aforementioned task, the agent can sequentially use the skills: harvest log 🪵, ..., craft furnace 🧱, harvest beef 🥩, place furnace 🧱, and craft cooked_beef 🍖. Each skill solves a simple sub-task in a shorter time horizon, with the necessary tools and conditions provided. For example, the skill 'craft cooked_beef 🍖' solves the task 'harvest cooked_beef 🍖 with beef 🥩, log 🪵, and placed furnace 🧱'. Once the agent acquires an abundant set of skills $S$, it can solve any complex task by decomposing it into a sequence of sub-tasks and executing the skills in order. Meanwhile, by reusing a skill to solve different tasks, the agent is much better in memory and learning efficiency. We assume that recent Large Language Models (LLMs), such as GPT-3.5 (Ouyang et al., 2022), contain high-level knowledge about skills in a task and their relationships in similar open-world scenarios.

To this end, we convert the goal of solving diverse and long-horizon tasks in Minecraft into building a hierarchical agent. At the low level, we train policies $\pi_s$ to learn all the skills $s \in S$, where $\pi_s$ takes as input the RGB image and some auxiliary information (compass, location, biome, etc.), then outputs an action. At the high level, we study planning methods to convert a task $\tau$ into a skill sequence $(s_{\tau,1}, s_{\tau,2}, \cdots)$.

### 2.2 Skills in Minecraft

Recent works mainly rely on imitation learning to learn Minecraft skills efficiently. In MineRL competition (Kanervisto et al., 2022), a human gameplay dataset is accessible along with the Minecraft environment. All of the top methods in competition use imitation learning to some degree, to learn useful behaviors in limited interactions. In VPT (Baker et al., 2022), a large policy model is pre-trained on a massive labeled dataset using behavior cloning. By fine-tuning on smaller datasets, policies are acquired for diverse skills.

However, without demonstration datasets, learning Minecraft skills with reinforcement learning (RL) is difficult. MineAgent (Fan et al., 2022) shows that PPO (Schulman et al., 2017) can only learn a small set of skills. PPO with sparse reward fails in 'milk a cow' and 'shear a sheep', though the distance between target mobs and the agent is set within 10 blocks. We argue that with the high dimensional state and action space, open-ended large world, and partial observation, exploration in Minecraft tasks is extremely difficult.

Table 1: Minecraft skill performance of imitation learning (behavior cloning with MineCLIP backbone, reported in (Cai et al., 2023)) versus reinforcement learning. *Better init.* means target entities are closer to the agent at initialization. The RL method for each task is trained with proper intrinsic rewards. All RL results are averaged on the last 100 training epochs and 3 training seeds.

| Skill | | | | | |
|---|---|---|---|---|---|
| Behavior Cloning | – | – | 0.25 | 0.27 | 0.16 |
| RL | 0.40±0.20 | 0.26±0.22 | 0.04±0.02 | 0.04±0.01 | 0.00±0.00 |
| RL (*better init.*) | 0.99±0.01 | 0.81±0.02 | 0.16±0.06 | 0.14±0.07 | 0.44±0.10 |

We conduct a study for RL to learn skills with different difficulties in Table 1. We observe that RL has comparable performance to imitation learning only when the task-relevant entities are initialized very close to the agent. Otherwise, RL performance decreases significantly. This motivates us to further divide skills into fine-grained skills. We propose a **Finding-skill** to provide a good initialization for other skills. For example, the skill of 'milk a cow' is decomposed into 'find a cow' and 'harvest milk_bucket'. After finding a cow nearby, 'harvest milk_bucket' can be accomplished by RL with acceptable sample efficiency. Thus, learning such fine-grained skills is easier for RL, and they together can still accomplish the original task.

## 3    Learning Basic Skills with Reinforcement Learning

Based on the discussion above, we propose three types of fine-grained basic skills, which can compose all Minecraft tasks.

- The Finding-skill: starts from any location, the agent explores to find a target and approaches the target. The target can be any block or entity that exists in the world.

- Manipulation-skills: given proper tools and the target in sight, the agent interacts with the target to obtain materials. These skills include diverse behaviors, like mining ores, killing mobs, and placing blocks.

- Crafting-skills: with requisite materials in the inventory and crafting table or furnace placed nearby, the agent crafts advanced materials or tools.

### 3.1    Learning to Find with a Hierarchical Policy

Finding items is a long-horizon difficult task for RL. To find an unseen tree on the plains, the agent should take thousands of steps to explore the world map as much as possible. A random policy fails to do such exploration, as shown in Appendix G. Also, it is too costly to train different policies for various target items. To simplify this problem, considering to explore on the world's surface only, we propose to train a target-free hierarchical policy to acquire the Finding-skill.

Figure 2 demonstrates the hierarchical policy for the Finding-skill. The high-level policy $\pi^H \left((x,y)^g | (x,y)_{0:t}\right)$ observes historical locations $(x,y)_{0:t}$ of the agent, and outputs a goal location $(x,y)^g$. It drives the low-level policy $\pi^L \left(a_t | o_t, (x,y)^g\right)$ to reach the goal location. We assume that target items are uniformly distributed on the world's surface. To maximize the chance to find diverse targets, the objective for the high-level policy is to maximize its reached area. We divide the world's surface into discrete grids, where each grid represents a $10 \times 10$ area. We use state count in the grids as the reward for the high-level policy. The low-level policy obtains the environmental observation $o_t$ and the goal location $(x,y)^g$ proposed by the high-level policy, and outputs an action $a_t$. We reward the low-level policy with the distance change to the goal location.

To train the hierarchical policy with acceptable sample complexity, we pre-train the low-level policy with randomly generated goal locations using DQN (Mnih et al., 2015), then train the high-level policy using PPO (Schulman et al., 2017) with the fixed low-level policy. During test, the goal of the Finding-skill is to

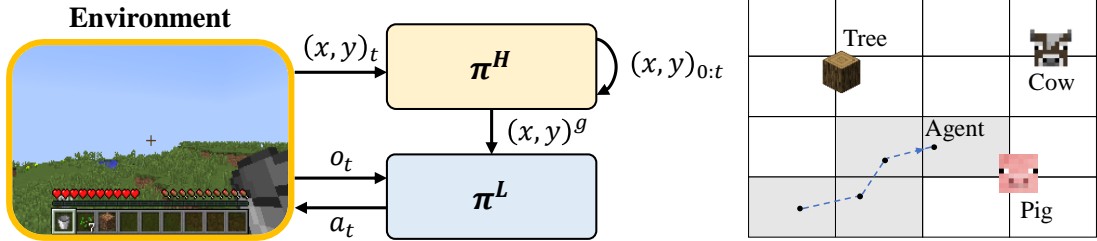

Figure 2: The proposed hierarchical policy for the Finding-skill. The high-level recurrent policy $\pi^H$ observes historical positions $(x,y)_{0:t}$ from the environment and generates a goal position $(x,y)^g$. The low-level policy $\pi^L$ is a goal-based policy to reach the goal position. The right figure shows a top view of the agent's exploration trajectory, where the walking paths of the low-level policy are shown in blue dotted lines, and the goal is changed by the high-level policy at each black spot. The high-level policy is optimized to maximize the state count in the grid world, which is shown in the grey background.

find a specific item required by the subsequent skill, as detailed in the structured skill information described in Section 4.1. The agent first explores the world with the hierarchical policy until the target item is detected in its lidar observations. Then, the agent executes the low-level policy conditioned on the detected target's location, to reach the target item. Though we use additional lidar information here, we believe that for other environments lacking this information, we can also implement the success detector for the Finding-skill with computer vision models such as vision-language models (Du et al., 2023).

## 3.2 Manipulation and Crafting

By executing the pre-trained Finding-skill, we can instantiate the manipulation tasks with requisite target items nearby, making the manipulation tasks much easier. To train the Manipulation-skills in Minecraft, we can either make a training environment with the target item initialized nearby or run the Finding-skill to reach a target item. For example, to train the skill 'harvest milk_bucket ', we can either spawn a cow close to the agent using the Minecraft built-in commands, or execute the Finding-skill until a cow is reached. The latter is similar in the idea to Go-Explore (Ecoffet et al., 2019), and is more suitable for other environments that do not have commands to initialize the target items nearby.

We adopt MineCLIP (Fan et al., 2022) to guide the agent with intrinsic rewards. The pre-trained MineCLIP model computes the CLIP reward based on the similarity between environmental observations (frames) and the language descriptions of the skill. We train the agent using PPO with self-imitation learning, to maximize a weighted sum of intrinsic rewards and extrinsic success (sparse) reward. Details for training basic skills can be found in Appendix C.

For the Crafting-skills, they can be executed with only a single action in MineDojo (Fan et al., 2022).

## 4 Solving Minecraft Tasks via Skill Planning

In this section, we present our skill planning method for solving diverse hard tasks. A skill graph is generated in advance with a Large Language Model (LLM), enabling searching for correct skill sequences on the fly.

### 4.1 Constructing Skill Graph with Large Language Models

A correct plan $(s_{\tau,1}, s_{\tau,2}, \cdots)$ for a task $\tau = (g, I)$ should satisfy two conditions. (1) For each $i$, $s_{\tau,i}$ is executable after $(s_{\tau,1}, \cdots, s_{\tau,i-1})$ are accomplished sequentially with initial condition $I$. (2) The target item $g$ is obtained after all the skills are accomplished sequentially, given initial condition $I$. To enable searching for such plans, we should be able to verify whether a plan is correct. Thus, we should know what condition is required and what is obtained for each skill. We define such information of skills in a structured format. As an example, information for skill 'crafting stone_pickaxe ' is:

```
stone_pickaxe {consume: {cobblestone: 3, stick: 2},
require: {crafting_table_nearby: 1}, obtain: {stone_pickaxe: 1}}
```

Each item in this format is also a skill. Regarding them as graph nodes, this format shows a graph structure between skill 'stone_pickaxe' and skills 'cobblestone', 'stick', 'crafting_table_nearby'. The directed edge from 'cobblestone' to 'stone_pickaxe' is represented as (3, 1, consume), showing the quantity relationship between parent and child, and that the parent item will be consumed during skill execution. In fact, in this format, all the basic skills in Minecraft construct a large directed acyclic graph with hundreds of nodes. The dashed box in Figure 1 shows a small part of this graph, where grey arrows denote 'consume' and red arrows denote 'require'.

To construct the skill graph, we generate structured information for all the skills by interacting with GPT-3.5 (Ouyang et al., 2022), a high-performance LLM. Since LLMs are trained on large-scale internet datasets, they obtain rich knowledge in the popular game Minecraft. In the prompt, we give a few demonstrations and explanations about the format, then ask the LLM to generate other skills information. Dialog with the LLM can be found in Appendix D.

### 4.2 Skill Search Algorithm

Our skill planning method is a depth-first search (DFS) algorithm on the skill graph. Given a task $\tau = (g, I)$, we start from the node $g$ and do DFS toward its parents, opposite to the edge directions. In this process, we maintain all the possessing items starting from $I$. Once conditions for the skill are satisfied or the skill node has no parent, we append this skill into the planned skill list and modify the maintained items according to the skill information. The resulting skill list is ensured to be executable and target-reaching.

To solve a long-horizon task, since the learned low-level skills are possible to fail, we alternate skill planning and skill execution until the episode terminates. After each skill execution, we update the agent's condition $I'$ based on its inventory and the last executed skill, and search for the next skill with $\tau' = (g, I')$.

We present the pseudocode for the iterative test process in Algorithm 1 and leave the skill search algorithm in Appendix B.

---

**Algorithm 1:** Process for solving a task.

**Input:** Task: $T = (g, I)$; Pre-trained skills: $\{\pi_s\}_{s \in S}$; Pre-generated skill graph: $G$; Skill search algorithm: *Search*.

**Output:** Task success.

$I' \leftarrow I$;
**while** episode not terminated **do**
    $(s_1, s_2, ...) \leftarrow Search(G, g, I')$;
    Execute $\pi_{s_1}$ for several steps;
    **if** *task success* **then**
        return **True**;
    $I' \leftarrow$ (inventory items, nearby items);
return **False**.

---

### 4.3 Method Summary

Here, we provide a concise summary of the training and test procedures for our agent. Initially, we use the LLM to generate information of all skills in Minecraft and construct a skill graph. Then, we acquire different skills with RL, involving two parts: (1) training the Finding-skill using hierarchical RL; (2) training each manipulation skill with RL, where we either employ the Finding-skill for effective task reset or directly create an environment where target items are spawned closely to the agent. Upon completion of these steps, we assemble the agent, integrating the skill graph, skill search algorithm, and the trained policies for each skill. Given any task, the agent solves it through iterative skill planning and execution.

## 5 Experiments

In this section, we evaluate and analyze our method with baselines and ablations in challenging Minecraft tasks. Section 5.1 introduces the implementation of basic skills. In Section 5.2, we introduce the setup for our evaluation task suite. In Section 5.3 and 5.4, we present the experimental results and analyze skill learning and planning respectively.

### 5.1 Training Basic Skills

To pre-train basic skills with RL, we use the environments of programmatic tasks in MineDojo (Fan et al., 2022). To train Manipulation-skills, for simplicity, we specify the environment that initializes target mobs or resources close to the agent. For the Go-Explore-like training method without specified environments discussed in Section 3.2, we present the results in Appendix H, which does not underperform the former.

For Manipulation-skills and the low-level policy of the Finding-skill, we adopt the policy architecture of MineAgent (Fan et al., 2022), which uses a fixed pre-trained MineCLIP image encoder and processes features using MLPs. To explore in a compact action space, we compress the original large action space into $12 \times 3$ discrete actions. For the high-level policy of the Finding-skill, which observes the agent's past locations, we use an LSTM policy and train it with truncated BPTT (Pleines et al., 2023). We pick the model with the highest success rate on the smoothed training curve for each skill, and fix these policies in all tasks. Implementation details can be found in Appendix C.

Note that Plan4MC totally takes 7M environment steps in training, and can unlock the iron pickaxe ⛏ in the Minecraft Tech Tree in test. This level of sample efficiency is conducive to conducting experiments on lab machines, in contrast to other existing RL methods (Hafner et al., 2023; Baker et al., 2022), which require hundreds of millions of time steps.

### 5.2 Task Setup

Based on MineDojo (Fan et al., 2022) programmatic tasks, we set up an evaluation benchmark consisting of four groups of diverse tasks: cutting trees 🪵 to craft primary items, mining cobblestones 🪨 to craft intermediate items, mining iron ores 🟫 to craft advanced items, and interacting with mobs 🐄 to harvest food and materials. Each task set has 10 tasks, adding up to a total of 40 tasks. With our settings of basic skills, these tasks require 25 planning steps on average and maximally 121 planning steps. We estimate the number of the required steps for each task with the sum of the steps of the initially planned skills and double this number to be the maximum episode length for the task, allowing skill executions to fail. The easiest tasks have 3000 maximum steps, while the hardest tasks have 12000. More details about task setup are listed in Appendix E.

To evaluate the success rate on each task, we average the results over 30 test episodes. In practice, for deploying and testing our agent, we can select the best RL policy for each skill across various training episodes and seeds. Due to the low simulation speed of the Minecraft game, we train each skill with a single seed and select the checkpoint with the highest success rates on smoothed training curves, aligning with practices in the field (Baker et al., 2022).

### 5.3 Skill Learning

We first analyze learning basic skills. While we propose three types of fine-grained basic skills, others directly learn more complicated and long-horizon skills. We introduce two baselines to study learning skills with RL.

**MineAgent (Fan et al., 2022).** Without decomposing tasks into basic skills, MineAgent solves tasks using PPO and self-imitation learning with the CLIP reward. For fairness, we train MineAgent in the test environment for each task. The training takes 7M environment steps, which is equal to the sum of environment steps we take for training all the basic skills. We average the success rate of trajectories in the last 100 training epochs (around 1M environment steps) to be its test success rate. Since MineAgent has no actions for crafting items, we hardcode the crafting actions into the training code. During trajectory

Table 2: Average success rates on four task sets of our method, all the baselines and ablation methods. Success rates on all the single tasks are listed in Appendix F.

| Task Set | Cut-Trees | Mine-Stones | Mine-Ores | Interact-Mobs |
|---|---|---|---|---|
| MineAgent | 0.003 | 0.026 | 0.000 | 0.171 |
| Plan4MC w/o Find-skill | 0.187 | 0.097 | 0.243 | 0.170 |
| LLM Planner | 0.260 | 0.067 | 0.030 | 0.247 |
| Plan4MC Zero-shot | 0.183 | 0.000 | 0.000 | 0.133 |
| Plan4MC 1/2-steps | 0.337 | 0.163 | 0.143 | 0.277 |
| **Plan4MC** | **0.417** | **0.293** | **0.267** | **0.320** |

collection, at each time step where the skill search algorithm returns a Crafting-skill, the corresponding crafting action will be executed. Note that, if we expand the action space for MineAgent rather than automatically execute crafting actions, the exploration will be much harder.

**Plan4MC w/o Find-skill.** None of the previous work decomposes a skill into executing the Finding-skill and Manipulation-skills. Instead, finding items and manipulations are done with a single skill. Plan4MC w/o Find-skill implements such a method. It skips the Finding-skill in the skill plans during test. Manipulation-skills take over the whole process of finding items and manipulating them.

Table 2 shows the test results for these methods. Plan4MC outperforms two baselines on the four task sets. MineAgent fails on the task sets of Cut-Trees, Mine-Stones and Mine-Ores, since taking many attacking actions continually to mine blocks in Minecraft is an exploration difficulty for RL on long-horizon tasks. On the contrary, MineAgent achieves performance comparable to Plan4MC's on some easier tasks 🪣🧊🥩🍖 in Interact-Mobs, which requires fewer environment steps and planning steps. Plan4MC w/o Find-skill consistently underperforms Plan4MC on all the tasks, showing that introducing the Finding-skill is beneficial for solving hard tasks with basic skills trained by RL. Because there is no Finding-skill in harvesting iron ores, their performance gap on Mine-Ores tasks is small.

To further study the Finding-skill, we present the success rate at each planning step in Figure 3 for three tasks. The curves of Plan4MC and Plan4MC w/o Find-skill have large drops at the Finding-skill. Especially, the success rates at finding cobblestones and logs decrease the most, because these items are harder to find in the environment compared to mobs. In these tasks, we compute the average success rate of Manipulation-Skills, conditioned on the skill before the last Finding-skill being accomplished. While Plan4MC has a conditional success rate of 0.40, Plan4MC w/o Find-skill decreases to 0.25, showing that solving sub-tasks with the additional Finding-skill is more effective.

As shown in Table 3, most Manipulation-skills have slightly lower success rates in test than in training, due to the domain gap between test and training environments. However, this decrease does not occur in skills 🪣🧊 that are trained with a large initial distance of mobs/items, as the pre-executed Finding-skill provide better initialization for Manipulation-skills during the test and thus the success rate may increase. In contrast, the success rates in the test without Finding-skill are significantly lower.

## 5.4 Skill Planning

For skill planning in open-ended worlds, recent works (Huang et al., 2022a;b; Brohan et al., 2023; Liang et al., 2022; Wang et al., 2023c) generate plans or sub-tasks with LLMs. We study these methods on our task sets and implement a best-performing baseline to compare with Plan4MC.

**LLM Planner.** We implement an LLM-based planner with GPT-3.5 (Ouyang et al., 2022), which proposes skill plans based on prompts including descriptions of tasks and observations. Similar to chain-of-thoughts prompting (Wei et al., 2022), we provide few-shot demonstrations with explanations to the planner at the initial planning step. In addition, we add several rules for planning into the prompt to fix common errors that the model encountered during test. At each subsequent planning step, the planner will encounter one

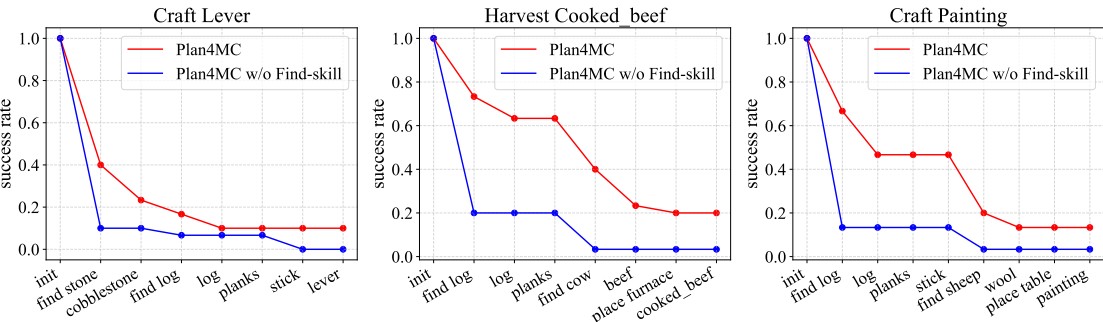

Figure 3: Success rates of Plan4MC with/without Finding-skill at each skill planning step, on three long-horizon tasks. We arrange the initially planned skill sequence on the horizontal axis and remove the repeated skills. The success rate of each skill represents the probability of successfully executing this skill at least once in a test episode. Specifically, the success rate is always 1 at task initialization, and the success rate of the last skill is equal to the task's success rate.

Table 3: Success rates of Manipulation-skills in training and test. *Training init. distance* is the maximum distance for mobs/items initialization in training skills. Note that in test, executing the Finding-skill will reach the target items within a distance of 3. *Training success rate* is averaged over 100 training epochs around the selected model's epoch. *Test success rate* is computed from the test rollouts of all the tasks, while *w/o Find* refers to Plan4MC w/o Find-skill.

| Manipulation-skills | Place | | | | | | |
|---|---|---|---|---|---|---|---|
| Training init. distance | -- | 10 | 10 | 2 | 2 | -- | -- |
| Training success rate | 0.98 | 0.50 | 0.27 | 0.21 | 0.30 | 0.56 | 0.47 |
| Test success rate | 0.77 | 0.71 | 0.26 | 0.27 | 0.16 | 0.33 | 0.26 |
| Test success rate (w/o Find) | 0.79 | 0.07 | 0.03 | 0.03 | 0.02 | 0.05 | 0.06 |

of the following cases: the proposed skill name is invalid, the skill is already done, skill execution succeeds, and skill execution fails. We carefully design language feedback for each case and ask the planner to re-plan based on inventory changes. For low-level skills, we use the same pre-trained skills as Plan4MC.

Also, we conduct ablations on our skill planning designs.

**Plan4MC Zero-shot.** This is a zero-shot variant of our interactive planning method, proposing a skill sequence at the beginning of each task only. The agent executes the planned skills sequentially until a skill fails or the environment terminates. This planner has no fault tolerance for skills execution.

**Plan4MC 1/2-steps.** In this ablation study, we half the test episode length and require the agent to solve tasks more efficiently.

Success rates for each method are listed in Table 2. We find that LLM Planner has comparable performance to Plan4MC on the task set of Interact-Mobs, where most tasks require less than 10 planning steps. In Mine-Stones and Mine-Ores tasks with long-horizon planning, the LLM planner is more likely to make mistakes, resulting in worse performance. The performance of Plan4MC Zero-shot is much worse than Plan4MC in all the tasks, since a success test episode requires accomplishing each skill in one trial. The decrease is related to the number of planning steps and skills success rates in Table 3.

As shown in Figure 4, Plan4MC 1/2-steps has close performance to Plan4MC at each planning step, while the Plan4MC Zero-shot fails after executing skills with low success rates. Plan4MC 1/2-steps has the least performance decrease to Plan4MC, showing that Plan4MC can solve tasks in a very limited episode length.

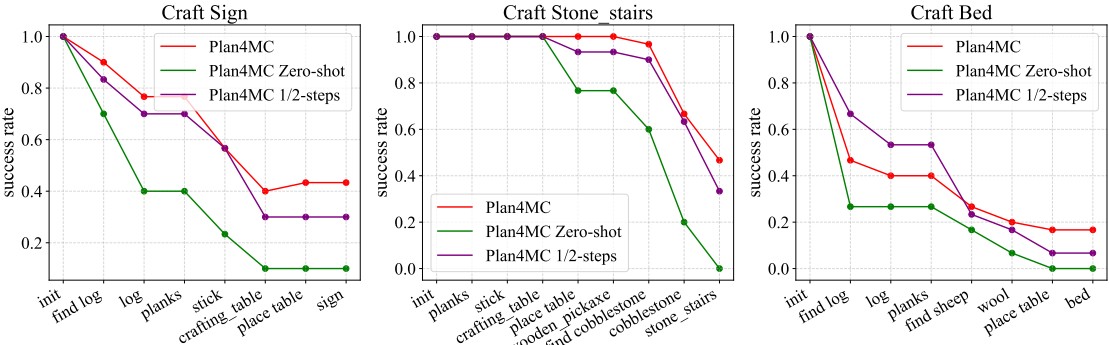

Figure 4: Success rates of Plan4MC compared with ablated skill planning methods at each planning step, on three long-horizon tasks. We arrange the initial planned skill sequence on the horizontal axis and remove the repeated skills. The success rate of each skill represents the probability of successfully executing this skill at least once in a test episode. Specifically, the success rate is always 1 at task initialization, and the success rate of the last skill is equal to the task's success rate.

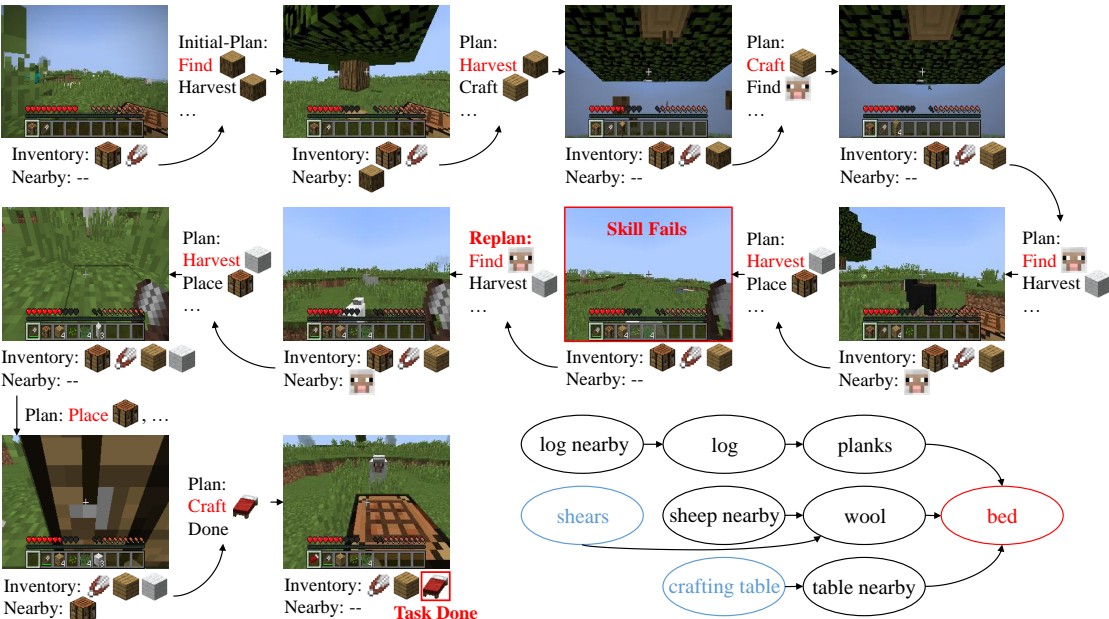

Figure 5: Demonstration of a planning and execution episode for the task 'craft a bed'. Following the direction of the arrows, the planner iteratively proposes the skill sequence based on the agent's state, and the policy executes the first skill. Though an execution for 'harvest wool' fails in the middle, the planner replans to 'find a sheep' again to fix this error, and finally completes the task. The lower right shows the skill graph for this task, where the red circle indicates the target and the blue circles indicate the initial items.

## 5.5 Pipeline Visualization

Here, we visually demonstrate the steps Plan4MC takes to solve a long-horizon task. Figure 5 shows the interactive planning and execution process for crafting a bed . With the interactive planning mechanism, Plan4MC allows execution failures of low-level policies and can revise the plan based on the agent state.

# 6   Related Work

**Agents in Minecraft.** In recent years, the popular open-world game Minecraft has become a challenging benchmark for testing AI agents. Early works tackle the ObtainDiamond challenge (Milani et al., 2020; Guss et al., 2021; Kanervisto et al., 2022) with hierarchical RL and imitation learning approaches (Milani et al., 2020; Skrynnik et al., 2021; Mao et al., 2022; Lin et al., 2022; Amiranashvili et al., 2020), where datasets of expert demonstrations are utilized to facilitate RL training. Recently, several works (Fan et al., 2022; Lifshitz et al., 2023; Baker et al., 2022; Yuan et al., 2024) leverage internet-scale video datasets to facilitate policy learning. However, they either can only solve simple short-term tasks or take a substantial number of environment steps to learn long-horizon tasks. Other works explore multi-task learning (Tessler et al., 2017; Kanitscheider et al., 2021; Cai et al., 2023; Nottingham et al., 2023), unsupervised skill discovery (Nieto et al., 2021), LLM-based planning (Wang et al., 2023c;a; Zhu et al., 2023) for Minecraft Agents. Our Plan4MC is the first demonstration-free, efficient RL agent in Minecraft to solve diverse long-horizon tasks including many challenging Tech Tree tasks.

**Learning Skills in Minecraft.** Acquiring skills is crucial for solving long-horizon tasks in Minecraft. Hierarchical approaches (Mao et al., 2022; Lin et al., 2022) in MineRL competition learn low-level skills with imitation learning. VPT (Baker et al., 2022) labels internet-scale datasets and pre-trains a behavior-cloning agent to initialize for diverse tasks. Recent works (Cai et al., 2023; Wang et al., 2023c; Nottingham et al., 2023; Lifshitz et al., 2023) learn skills based on VPT. Without expert demonstrations, MineAgent (Fan et al., 2022) and CLIP4MC (Ding et al., 2023) learn skills with RL and vision-language rewards. But they can only acquire a small set of skills. Unsupervised skill discovery (Nieto et al., 2021) learns skills that only produce different navigation behaviors. In our work, to enable RL to acquire diverse skills, we introduce the Finding-skill to provide state initialization and effectively learn fine-grained basic skills with intrinsic rewards.

**Planning with Large Language Models.** With the rapid progress of LLMs (Ouyang et al., 2022; Chowdhery et al., 2022), many works study LLMs as planners in open-ended worlds. To ground language models, SayCan (Brohan et al., 2023) combines LLMs with skill affordances to produce feasible plans, Translation LMs (Huang et al., 2022a) selects demonstrations to prompt LLMs, and LID (Li et al., 2022) finetunes language models with tokenized interaction data. LiFT (Nam et al., 2023) and BOSS (Zhang et al., 2023) uses LLMs to facilitate skill training. Other works study interactive planning for error correction. Inner Monologue (Huang et al., 2022b) proposes environment feedback to the planner. DEPS (Wang et al., 2023c) introduces descriptor, explainer, and selector to generate plans by LLMs. Many works (Wang et al., 2023a; Zhu et al., 2023; Wang et al., 2023b; Zhao et al., 2023) study LLM-based agents in Minecraft where LLMs are high-level planners and the low-level controllers are manually designed (PrismarineJS, 2013). In our work, we leverage the LLM to generate a skill graph and introduce a skill search algorithm to eliminate planning mistakes.

# 7   Conclusion and Discussion

In this paper, we propose a framework to solve diverse long-horizon open-world tasks with reinforcement learning and planning. To tackle the exploration and sample efficiency issues, we propose to learn fine-grained basic skills with RL and introduce a general Finding-skill to provide good environment initialization for skill learning. In Minecraft, we design a graph-based planner, taking advantage of the prior knowledge in LLMs and the planning accuracy of the skill search algorithm. Experiments on 40 challenging Minecraft tasks verify the advantages of Plan4MC over various baselines.

Though we implement Plan4MC in Minecraft, our method is extendable to other similar open-world environments and draws insights on building multi-task learning systems. We leave the detailed discussion in Appendix I.

## 8    Limitations

When implementing the Finding-skill in Minecraft, we operate under the assumption that target items are uniformly distributed and the world is divided into grids, which may not be realistic and effective assumptions. Furthermore, the Finding-skill lacks goal awareness during exploration, rendering the goal-reaching policy sub-optimal. Future efforts should focus on relaxing these assumptions and developing a goal-aware policy.

Correcting the outputs of the LLM in building the skill graph poses a challenge for future research, especially when the LLM lacks specific domain knowledge. In Minecraft, GPT-3.5 makes two kinds of mistakes. The first is rule and format errors. For example, the LLM generates "mobs" instead of "mobs_nearby" as required in our skills. The latter is related to Minecraft domain knowledge. For example, to craft shears, the LLM says that a crafting_table_nearby is required but it is not necessary in fact. Using external knowledge such as Wiki documents is a promising approach to correct such errors in the LLM outputs. Similar ideas appear in recent work (Zhu et al., 2023; Wu et al., 2023) that improve LLM results using domain-specific databases. Another alternative approach is to correct the skill mistakes leveraging environmental feedback, which is studied in concurrent work (Wang et al., 2023a; Zhu et al., 2023).

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

## A  Video Demonstration

Figure 6 shows the key frames of Plan4MC solving the challenging Tech Tree task of crafting an iron pickaxe with bare hands.

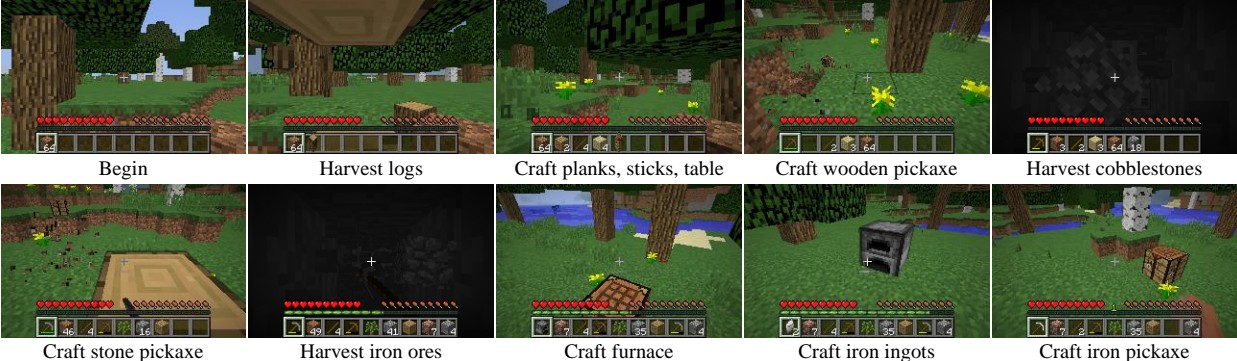

Figure 6: A playing episode of Plan4MC for crafting iron pickaxe **with bare hands**. This is a challenging task in Minecraft Tech Tree, which requires 16 different basic skills and 117 steps in the initial plan.

## B  Algorithms

We present our algorithm sketches for skill planning here.

---

**Algorithm 2:** DFS.

---

**Input:** Pre-generated skill graph: $G$; Target item: $g$; Target item quantity: $n$;
*Global variables*: possessing items $I$ and skill sequence $S$.

**for** $g'$ *in parents*$(G, g)$ **do**
  $n_{g'}, n_g, consume \leftarrow\ <g', g>$;
  $n_{g'}^{todo} \leftarrow n_{g'}$;
  **if** *(quantity of $g'$ in $I$)* $> n_{g'}$ **then**
    Decrease $g'$ quantity with $n_{g'}$ in $I$, if *consume*;
  **else**
    $n_{g'}^{todo} \leftarrow n_{g'}^{todo} -$ (quantity of $g'$ in $I$);
  **while** $n_{g'}^{todo} > 0$ **do**
    DFS$(G, g', n_{g'}^{todo}, I, S)$;
    **if** $g'$ *is not Crafting-skill* **then**
      Remove all nearby items in $I$;
    $n_{g'}^{obtain} \leftarrow$ (quantity of $g'$ obtained after executing skill $g'$);
    **if** $n_{g'}^{obtain} > n_{g'}^{todo}$ **then**
      Increase $g'$ quantity with $n_{g'}^{obtain} - n_{g'}^{todo}$ in $I$;
    Increase other obtained items after executing skill $g'$ in $I$;
    $n_{g'}^{todo} \leftarrow n_{g'}^{todo} - n_{g'}^{obtain}$;
Append skill $g$ to $S$.

---

---

**Algorithm 3:** Skill search algorithm.

---

**Input:** Pre-generated skill graph: $G$; Target item: $g$; Initial items: $I$.
**Output:** Skill sequence: $(s_1, s_2, ...)$.

$S' \leftarrow ()$;
$I' \leftarrow I$;
DFS$(G, g, 1, I', S')$;
return $S'$.

---

## C  Details in Training Basic Skills

Table 4 shows the environment and algorithm configurations for training basic skills. Except for the skill of mining ⬛🟩 whose breaking speed multiplier in the simulator is set to 10, all the skills are trained using the unmodified MineDojo simulator.

Though the MineCLIP reward improves the learning of many skills, it is still not enough to encourage some complicated behaviors. In combat 🐄🐷, we introduce distance reward and attack reward to further encourage the agent to chase and attack the mobs. In mining 🟫⬛, we introduce distance reward to keep the agent close to the target blocks. To mine underground ores ⬛🟩, we add depth reward to encourage the agent to mine deeper and then go back to the ground. These item-based intrinsic rewards are easy to implement for all the items and are also applicable in many other open-world environments like robotics. Intrinsic rewards are implemented as follows.

**State count.** The high-level recurrent policy for the Finding-skill optimizes the visited area in a $110 \times 110$ square, where the agent's spawn location is at the center. We divide the square into $11 \times 11$ grids and keep a visitation flag for each grid. Once the agent walks into an unvisited grid, it receives $+1$ state count reward.

**Goal navigation.** The low-level policy for the Finding-skill is encouraged to reach the goal position. The goal location is randomly sampled in 4 directions at a distance of 10 from the agent. To get closer to the goal, we compute the distance change between the goal and the agent: $r_d = -(d_t - d_{t-1})$, where $d_t$ is the distance on the plane coordinates at time step $t$. Additionally, to encourage the agent to look in its walking direction, we add rewards to regularize the agent's yaw and pitch angles: $r_{yaw} = yaw \cdot g, r_{pitch} = cos(pitch)$, where $g$ is the goal direction. The total reward is:

$$r = r_{yaw} + r_{pitch} + 10 * r_d. \tag{1}$$

**CLIP reward.** This reward encourages the agent to produce behaviors that match the task prompt. We sample 31 task prompts among all the MineDojo programmatic tasks as negative samples. The pre-trained MineCLIP (Fan et al., 2022) model computes the similarities between features of the past 16 frames and prompts. We compute the probability that the frames are most similar to the task prompt: $p = [\text{softmax}\left(S\left(f_v, f_l\right), \{S\left(f_v, f_{l^-}\right)\}_{l^-}\right)]_0$, where $f_v, f_l$ are video features and prompt features, $l$ is the task prompt, and $l^-$ are negative prompts. The CLIP reward is:

$$r_{\text{CLIP}} = \max\left\{p - \frac{1}{32}, 0\right\}. \tag{2}$$

**Distance.** The distance reward provides dense reward signals to reach the target items. For combat tasks, the agent gets a distance reward when the distance is closer than the minimal distance in history:

$$r_{distance} = \max\left\{\min_{t'<t} d_{t'} - d_t, 0\right\}. \tag{3}$$

Table 4: Training configurations for all the basic skills. *Max Steps* is the maximal episode length. *Training Steps* shows the environment steps cost for training each skill. *Init.* shows the maximal distance to spawn mobs at environment reset. The high-level policy and low-level policy for the Finding-skill are listed in two lines.

| Skill | Max Steps | Method | Intrinsic Reward | Training Steps | Biome | Init. |
|---|---|---|---|---|---|---|
| Find | high: 40 | PPO | state count | 1M | plains | -- |
|  | low: 50 | DQN | goal navigation | 0.5M |  |  |
| Place | 200 | PPO | CLIP reward | 0.3M | -- | -- |
| Harvest | 200 | PPO | CLIP reward | 1M | plains | 10 |
| Harvest | 200 | PPO | CLIP reward | 1M | plains | 10 |
| Combat | 400 | PPO | CLIP, distance, attack | 1M | plains | 2 |
| Combat | 400 | PPO | CLIP, distance, attack | 1M | plains | 2 |
| Harvest | 500 | PPO | distance | 0.5M | forest | -- |
| Harvest | 1000 | PPO | distance | 0.3M | hills | -- |
| Mine | 50 | PPO | depth | 0.4M | forest | -- |
| Craft | 1 | -- | -- | 0 | -- | -- |

For mining tasks, since the agent should stay close to the block for many time steps, we modify the distance reward to encourage keeping a small distance:

$$r_{distance} = \begin{cases} d_{t-1} - d_t, & 1.5 \leq d_t \leq +\infty \\ 2, & d_t < 1.5 \\ -2, & d_t = +\infty, \end{cases} \tag{4}$$

where $d_t$ is the distance between the agent and the target item at time step $t$, which is detected by lidar rays in the simulator.

**Attack.** For combat tasks, we reward the agent for attacking the target mobs. We use the tool's durability information to detect valid attacks and use lidar rays to detect the target mob. The attack reward is:

$$r_{attack} = \begin{cases} 90, & \text{if valid attack and the target at center} \\ 1, & \text{if valid attack but the target not at center} \\ 0, & \text{otherwise.} \end{cases} \tag{5}$$

**Depth.** For mining tasks, the agent should dig down first, then go back to the ground. We use the y-axis to calculate the change of the agent's depth, and use the depth reward to encourage such behaviors. To train the dig-down policy, the depth reward is:

$$r_{down} = \max \left\{ \min_{t' < t} y_{t'} - y_t, 0 \right\}. \tag{6}$$

To train the go-back policy, the depth reward is:

$$r_{up} = \max \left\{ y_t - \max_{t' < t} y_{t'}, 0 \right\}. \tag{7}$$

For each Manipulation-skill, we use a linear combination of intrinsic reward and extrinsic success reward to train the policy.

It takes one day on a single TITAN Xp GPU to train each skill for 1M environment steps. Table 5 shows our selected basic skill policies for downstream tasks. Since the Finding-skill and the Mining skill has no success rate during training, we pick the models with the highest returns on the smoothed training curves. For other skills, we pick the models with the highest success rates on the smoothed training curves.

Table 5: Information for all the selected basic skill policies. *Success Rate* is the success rate of the selected policy on the smoothed training curve.

| Skill | Parameters | Execute Steps | Success Rate |
|---|---|---|---|
| Find | 0.9M | 1000 | -- |
| Place | 2.0M | 200 | 0.98 |
| Harvest | 2.0M | 200 | 0.50 |
| Harvest | 2.0M | 200 | 0.27 |
| Combat | 2.0M | 400 | 0.21 |
| Combat | 2.0M | 400 | 0.30 |
| Harvest | 2.0M | 500 | 0.56 |
| Harvest | 2.0M | 200 | 0.47 |
| Mine | 4.0M | 1000 | – |
| Craft | 0 | 1 | 1.00 |

## D  LLM Prompt Design

### D.1  Prompt for Skill Graph Construction

To construct the skill graph, we use GPT-3.5 to generate skill information showing relationships between skills. For both Manipulation-skills and Crafting-skills, we demonstrate several skills, explain the format, and then ask the LLM to generate all the skills in the same format.

Listing 1: Prompt for generating some Manipulation-skills and the LLM's response. We only modify the last querying sentence in the prompt to generate all the skills.

```
Prompt:
I am playing the game Minecraft. I define some basic skills, like attack something, collect
    something and place something nearby. I list the skills in a special format.
As an example:
furnace_nearby:
  consume:
    'furnace': 1
  require:
  equip: ['furnace']
  obtain:
    'furnace_nearby': 1
To understand this skill line by line: the skill is to get a furnace_nearby. 'consume' means
    things will be consumed or killed. In this skill, furnace*1 will be consumed. 'require
    ' means things are needed but will not be consumed. In this skill, nothing else is
    required. We should equip furnace to the first slot. If you do not have to equip
    anything, write 'equip: []'. Finally, we will obtain furnace_nearby*1.
Another example:
cobblestone:
  consume:
    'cobblestone_nearby': 1
  require:
    'wooden_pickaxe': 1
  equip: ['wooden_pickaxe']
  obtain:
    'cobblestone': 1
To understand: to mine a cobblestone, we will consume a nearby cobblestone. A wooden_pickaxe
    is required and should be equipped, but will not be consumed.
Now you understand the rule of this format. Please help me generate the following skills:
crafting_table_nearby, wool, beef, diamond

LLM:
Here are the generated skills in the specified format:
```

```
crafting_table_nearby:
consume:
'crafting_table': 1
require:
equip: ['crafting_table']
obtain:
'crafting_table_nearby': 1

wool:
consume:
'sheep_nearby': 1
require:
'shears': 1
equip: ['shears']
obtain:
'wool': 1

beef:
consume:
'cow_nearby': 1
require:
equip: []
obtain:
'beef': 1

diamond:
consume:
'diamond_ore_nearby': 1
require:
'iron_pickaxe': 1
equip: ['iron_pickaxe']
obtain:
'diamond': 1
```

Listing 2: Prompt for generating some Crafting-skills and the LLM's response. We only modify the last querying sentence in the prompt to generate all the skills.

**Prompt:**

```
I am playing the game Minecraft. I list all the crafting skills in this game in a special
    format.
As an example:
bed:
  consume:
    'planks': 3
    'wool': 3
  require:
    'crafting_table_nearby': 1
  equip: []
  obtain:
    'bed': 1
To understand this skill line by line: the skill is to craft a bed. We will consume planks*3
     and wool*3. To craft bed, a crafting_table_nearby is necessary and must be in the '
    require:' line.
For crafting skills the equipment is always []. Finally, we will obtain bed*1.
Another example:
cooked_beef:
  consume:
    'planks': 1
    'beef': 1
  require:
    'furnace_nearby': 1
  equip: []
  obtain:
    'cooked_beef': 1
Because we require a furnace_nearby. The fuel, which is always 'planks'*1, should be added
    to the 'consume' items.
```

```
For the 'require:' line, you should know whether the skill needs crafting_table_nearby,
    furnace_nearby, or nothing. If require nothing, just don't write anything in this line.
Now you understand the rule of this format. Please help me generate the following skills:
stick, chest, furnace,item_frame

LLM:
Sure, here are the crafting skills for the items you requested:

stick:
consume:
'planks': 2
require:
equip: []
obtain:
'stick': 4

chest:
consume:
'planks': 8
require:
'crafting_table_nearby': 1
equip: []
obtain:
'chest': 1

furnace:
consume:
'cobblestone': 8
require:
'crafting_table_nearby': 1
equip: []
obtain:
'furnace': 1

item_frame:
consume:
'stick': 8
'leather': 1
require:
'crafting_table_nearby': 1
equip: []
obtain:
'item_frame': 1
```

All the Finding-skill have the same format:

```
(item)_nearby:
    consume:
    require:
    equip: []
    obtain:
        (item)_nearby: 1
```

In our experiments, after generating all the skills, we check and correct them according to the Minecraft database. Among 55 skills, we find 6 mistakes in total.

## D.2 Prompt for the LLM Planner Baseline

Here, we present the prompt for the LLM Planner baseline. Designs of in-context learning examples, rules, and feedback mechanisms are carefully tuned.

Listing 3: Prompt for the LLM Planner baseline.

```
################## Main Prompt:
```

```
I am playing Minecraft. Based on my inventory, you need to decompose a hard task into basic
    skills, and tell me which basic skill to do next. Basic skills are labeled with "'" (for
     example, 'log' is a basic skill, and log is a Minecraft block).
Here are rules of the decomposition:
    1. Most basic skills are named by Minecraft items and blocks. Skills named by natural
        things mean to get something, e.g. 'log', 'stone', 'iron_ore'. Skills named by
        artificial things mean to craft or heat to get them, e.g. 'planks', 'stick', '
        iron_ingot'.
    2. Before we use crafting_table to craft, we need to prepare 1 crafting_table and enough
         materials in our inventory. For example, the task is craft_iron_bar, firstly we
        need to prepare 6 iron_ingot (as materials) and 1 crafting_table (as tools) in
        inventory.
    3. After preparing 1 crafting_table and enough materials, do 'place_crafting_table'
        first, then we can craft. For example, the task is craft_furnace, and we've prepared
         8 cobblestone (as materials) and 1 crafting_table (as tools) in inventory, we need
        to do 'place_crafting_table' and get a "crafting_table_nearby", then do 'furnace'.
    4. Before we use furnace to heat, we need to prepare 1 furnace and enough materials. For
         example, the task is craft_iron_ingot, we need to prepare 1 furnace (as tools) and
        1 iron_ore (as materials) in our inventory first.
    5. After preparing 1 furnace and enough materials for the task, do 'place_furnace', then
         we use it. For example, the task is craft_iron_ingot, and we've prepared 1 iron_ore
         (as materials) and 1 furnace (as tools),we need to do 'place_furnace' and get a "
        furnace_nearby" and then do 'iron_ingot'.
    6. A tip: Always prepare materials firstly, and build tools secondly so that we will not
        waste the tools.

Here are some examples, "Inventory" is our Minecraft inventory, please pay attention to it.
    "Task" is the name of a hard task. "Next skill" is the next basic skill I need to do. "
    Sequence" is the correct decomposition of the hard task. "Explanation" is the
    explanation of the sequence.

    Inventory: 5 stone_pickaxe, 1 log
    Task: craft_crafting_table
    Sequence: ['planks', 'crafting_table']
    Next skill: 'planks'
    Explanation: The task is craft_crafting_table, we need 4 planks as materials. We have 1
        log in inventory, therefore we don't need to do 'log', just do 'planks' to get 4
        planks. After that, we will have 4 planks in inventory, then we do 'crafting_table'
        to consume 4 planks and craft 1 crafting_table.

    Inventory: 5 stone_pickaxe
    Task: craft_iron_hoe
    Sequence: ['iron_ore', 'iron_ore', 'log', 'planks', 'crafting_table', '
        place_crafting_table', 'furnace', 'place_furnace', 'iron_ingot', 'iron_ingot', 'log
        ', 'planks', 'stick', 'log', 'planks', 'crafting_table', 'place_crafting_table', '
        iron_hoe']
    Next skill: 'iron_ore'
    Explanation: To craft iron_hoe, we need 2 iron_ingot and 2 stick (as materials), 1
        crafting_table (as tools). We have no materials in inventory, therefore we prepare
        iron_ingot firstly, prepare stick secondly, and prepare crafting_table lastly. To
        get iron_ingot, So the next basic skill is 'iron_ore'. We use 'iron_ore' to get 1
        iron_ore.

    Inventory: 4 stone_pickaxe, 31 cobblestone, 6 iron_ore
    Task: craft_iron_bar
    Sequence: ['log', 'planks', 'crafting_table', 'place_crafting_table', 'furnace', '
        place_furnace', 'iron_ingot', 'iron_ingot', 'iron_ingot', 'iron_ingot', 'iron_ingot
        ', 'log', 'planks', 'crafting_table', 'place_crafting_table', 'iron_bar']
    Next skill: 'log'
    Explanation: To craft iron_bar, we need to prepare 6 iron_ingot (as materials) firstly
        and 1 crafting_table (as tools) secondly. To get 6 iron_ingot, we need 6 iron_ore (
        as materials) and 1 furnace (as tools), and we already have enough iron_ore in
        inventory, so we need to build 1 furnace. To get 1 furnace, we need 8 cobblestone (
        as materials) and 1 crafting_table (as tools), and we already have enough
        cobblestone, so we need to build a crafting_table. By 'log', 'planks' and '
        crafting_table' we can build it, therefore the next basic skill is 'log'.

Here are examples without explanation:
```

```
    Inventory: 4 stone_pickaxe , 69 cobblestone , 3 iron_ore , 1 crafting_table
    Task: craft_bucket
    Sequence: [ 'place_crafting_table ', 'furnace ', 'place_furnace ', 'iron_ingot ', '
        iron_ingot ', 'iron_ingot ', 'log ', 'planks ', 'crafting_table ', 'place_crafting_table
        ', 'bucket ']
    Next skill: 'place_crafting_table '

    Inventory: 1 crafting_table_nearby , 2 stone_pickaxe , 56 cobblestone , 5 iron_ore
    Task: craft_iron_helmet
    Sequence: ['furnace ', 'place_furnace ', 'iron_ingot ', 'iron_ingot ', 'iron_ingot ', '
        iron_ingot ', 'iron_ingot ', 'log ', 'planks ', 'crafting_table ', 'place_crafting_table
        ', 'iron_helmet ']
    Next skill: 'furnace '

    Inventory: 1 furnace_nearby , 4 stone_pickaxe , 47 cobblestone , 2 iron_ore
    Task: craft_iron_hoe
    Sequence: ['iron_ingot ', 'iron_ingot ', 'log ', 'planks ', 'stick ', 'log ', 'planks ', '
        crafting_table ', 'place_crafting_table ', 'iron_hoe ']
    Next skill: 'iron_ingot '

Here are examples without explanation and sequence :

    Inventory: 1 furnace_nearby , 3 stone_pickaxe , 121 cobblestone , 2 iron_ore , 4 iron_ingot
    Task: craft_iron_bar
    Next skill: 'iron_ingot '

    Inventory: 4 stone_pickaxe , 33 cobblestone
    Task: craft_iron_ingot
    Next skill: 'iron_ore '

    Inventory: 4 stone_pickaxe , 39 cobblestone , 4 iron_ore , 1 log
    Task: craft_iron_shoe
    Next skill: 'planks '

Here 're my initial inventory and a hard task :
    Inventory: INVENTORY (with nearby TOOL)
    Task: TASK
    Next skill: ?
And I have done the basic skill 'SKILL ' and I have a nearby TOOL now .
I wonder how to do task: TASK .
Based on my inventory and a nearby TOOL, please tell me the next basic skill I need to do (
    as the format:" Next skill:") and your explanation .

################## Feedback and Self-Reflection :

#### Skill execution failure :
OK, I get your answer :
    Next skill: 'SKILL '

But I have failed to do the basic skill 'SKILL '.

Here 's my inventory: INVENTORY2

Based on my inventory , please tell me do I lack materials to do the skill 'SKILL '?
If not, please tell me do I lack tools in my inventory ?
If not either , please tell me do I need to place crafting_table ? Or do I need to place
    furnace ?
Now please tell me the next basic skill I need to do (as the format:" Next skill:") and your
     explanation .

#### Skill planning error :
OK, I get your answer :
    Next skill: 'SKILL '
And here 's my inventory: INVENTORY2
```

```
But 'SKILL' is not a basic skill.
Can 'SKILL' be decomposed? Or change an another basic skill?
I wonder how to do task: TASK.
Based on my new inventory, please tell me the next basic skill I need to do (as the format:"
    Next skill:") and your explanation.
```

#### Redundant skill planning:

```
OK, I get your answer:
    Next skill: 'SKILL'

But I have already gotten SKILL in my inventory and maybe we don't need to do it.
My inventory: INVENTORY2
I wonder how to do task: TASK.
Based on my new inventory, please tell me the next basic skill I need to do (as the format:"
    Next skill:") and your explanation.
```

#### Skill name error:

```
Sorry, I can't understand your answer.
And here's my inventory: INVENTORY2

I wonder how to do task: TASK.
Based on my new inventory, please tell me the next basic skill I need to do (as the format:"
    Next skill:") and your explanation.
```

#### Missing crafting-table:

```
OK, I get your answer:
    Next skill: 'SKILL'

But I don't have the crafting_table/furnace to place in my inventory and maybe we need to
    build it firstly.
My inventory: INVENTORY2
I wonder how to do task: TASK.
Based on my new inventory, please tell me the next basic skill I need to do (as the format:"
    Next skill:") and your explanation.
```

#### Skill success:

```
OK, I get your answer:
    Next skill: 'SKILL'

Now I have done a basic skill 'SKILL' and there're changes in my inventory.
Old inventory: INVENTORY1
New inventory: INVENTORY2

I wonder how to do task: TASK.
Based on my new inventory, please tell me the next basic skill I need to do (as the format:"
    Next skill:") and your explanation.
```

## E  Task Setup

Table 6, 7 lists settings for 40 evaluation tasks. To make sure the agent is spawned in an unseen environment in each test episode, we randomly transport the agent with a maximum distance of 500 at environment reset. For tasks involving interacting with mobs, we spawn cows and sheep with a maximum distance of 30, which is much larger than the spawning distance in training basic skills. For the Mine-Ores task set, we set the breaking speed multiplier to 10. For the other three task sets, we use the default simulator.

Table 6: Settings for Cut-Trees and Mine-Stones tasks. *Initial Tools* are provided in the inventory at each episode beginning. *Involved Skills* is the least number of basic skills the agent should master to accomplish the task. *Planning Steps* is the number of basic skills to be executed sequentially in the initial plans.

| Task Icon | Target Name | Initial Tools | Biome | Max Steps | Involved Skills | Planning Steps |
|---|---|---|---|---|---|---|
| | stick | -- | plains | 3000 | 4 | 4 |
| | crafting_table_ nearby | -- | plains | 3000 | 5 | 5 |
| | bowl | -- | forest | 3000 | 6 | 9 |
| | chest | -- | forest | 3000 | 6 | 12 |
| | trap_door | -- | forest | 3000 | 6 | 12 |
| | sign | -- | forest | 3000 | 7 | 13 |
| | wooden_shovel | -- | forest | 3000 | 7 | 10 |
| | wooden_sword | -- | forest | 3000 | 7 | 10 |
| | wooden_axe | -- | forest | 3000 | 7 | 13 |
| | wooden_pickaxe | -- | forest | 3000 | 7 | 13 |
| | furnace_nearby | *10 | hills | 5000 | 9 | 28 |
| | stone_stairs | *10 | hills | 5000 | 8 | 23 |
| | stone_slab | *10 | hills | 3000 | 8 | 17 |
| | cobblestone_wall | *10 | hills | 5000 | 8 | 23 |
| | lever | | forest_hills | 5000 | 7 | 7 |
| | torch | *10 | hills | 5000 | 11 | 30 |
| | stone_shovel | | forest_hills | 10000 | 9 | 12 |
| | stone_sword | | forest_hills | 10000 | 9 | 14 |
| | stone_axe | | forest_hills | 10000 | 9 | 16 |
| | stone_pickaxe | | forest_hills | 10000 | 9 | 16 |

## F  Experiment Results for All the Tasks

Table 8 shows the success rates of all the methods in all the tasks, grouped in 4 task sets.

Table 7: Settings for Mine-Ores and Interact-Mobs tasks. *Initial Tools* are provided in the inventory at each episode beginning. *Involved Skills* is the least number of basic skills the agent should master to accomplish the task. *Planning Steps* is the number of basic skills to be executed sequentially in the initial plans.

| Task Icon | Target Name | Initial Tools | Biome | Max Steps | Involved Skills | Planning Steps |
|---|---|---|---|---|---|---|
|  | iron_ingot | *5, *64 | forest | 8000 | 12 | 30 |
|  | tripwire_hook | *5, *64 | forest | 8000 | 14 | 35 |
|  | heavy_weighted_ pressure_plate | *5, *64 | forest | 10000 | 13 | 61 |
|  | shears | *5, *64 | forest | 10000 | 13 | 61 |
|  | bucket | *5, *64 | forest | 12000 | 13 | 91 |
|  | iron_trapdoor | *5, *64 | forest | 12000 | 13 | 121 |
|  | iron_shovel | *5, *64 | forest | 8000 | 14 | 35 |
|  | iron_sword | *5, *64 | forest | 10000 | 14 | 65 |
|  | iron_axe | *5, *64 | forest | 12000 | 14 | 95 |
|  | iron_pickaxe | *5, *64 | forest | 12000 | 14 | 95 |
|  | milk_bucket | , *3 | plains | 3000 | 4 | 4 |
|  | wool | , *2 | plains | 3000 | 3 | 3 |
|  | beef |  | plains | 3000 | 2 | 2 |
|  | mutton |  | plains | 3000 | 2 | 2 |
|  | bed | ,  | plains | 10000 | 7 | 11 |
|  | painting | ,  | plains | 10000 | 8 | 9 |
|  | carpet |  | plains | 3000 | 3 | 5 |
|  | item_frame | ,  | plains | 10000 | 8 | 9 |
|  | cooked_beef | ,  | plains | 10000 | 7 | 7 |
|  | cooked_mutton | ,  | plains | 10000 | 7 | 7 |

Table 8: Success rates in all the tasks. Each task is tested for 30 episodes, set with the same random seeds across different methods.

| Task | MineAgent | Plan4MC w/o Find-skill | Interactive LLM | Plan4MC Zero-shot | Plan4MC 1/2-steps | Plan4MC |
|---|---|---|---|---|---|---|
| | 0.00 | 0.03 | 0.30 | 0.27 | 0.30 | 0.30 |
| | 0.03 | 0.07 | 0.17 | 0.27 | 0.20 | 0.30 |
| | 0.00 | 0.40 | 0.07 | 0.27 | 0.57 | 0.47 |
| | 0.00 | 0.23 | 0.00 | 0.07 | 0.10 | 0.23 |
| | 0.00 | 0.07 | 0.03 | 0.20 | 0.27 | 0.37 |
| | 0.00 | 0.07 | 0.00 | 0.10 | 0.30 | 0.43 |
| | 0.00 | 0.37 | 0.73 | 0.23 | 0.50 | 0.70 |
| | 0.00 | 0.33 | 0.63 | 0.30 | 0.60 | 0.47 |
| | 0.00 | 0.23 | 0.47 | 0.13 | 0.27 | 0.37 |
| | 0.00 | 0.07 | 0.20 | 0.00 | 0.27 | 0.53 |
| | 0.00 | 0.17 | 0.00 | 0.00 | 0.13 | 0.37 |
| | 0.00 | 0.30 | 0.20 | 0.00 | 0.33 | 0.47 |
| | 0.00 | 0.20 | 0.03 | 0.00 | 0.37 | 0.53 |
| | 0.21 | 0.13 | 0.13 | 0.00 | 0.33 | 0.57 |
| | 0.00 | 0.00 | 0.00 | 0.00 | 0.10 | 0.10 |
| | 0.05 | 0.10 | 0.00 | 0.00 | 0.17 | 0.37 |
| | 0.00 | 0.00 | 0.10 | 0.00 | 0.03 | 0.20 |
| | 0.00 | 0.07 | 0.13 | 0.00 | 0.07 | 0.10 |
| | 0.00 | 0.00 | 0.07 | 0.00 | 0.10 | 0.07 |
| | 0.00 | 0.00 | 0.00 | 0.00 | 0.00 | 0.17 |
| | 0.00 | 0.53 | 0.20 | 0.00 | 0.30 | 0.47 |
| | 0.00 | 0.27 | 0.00 | 0.00 | 0.27 | 0.33 |
| | 0.00 | 0.37 | 0.00 | 0.00 | 0.13 | 0.30 |
| | 0.00 | 0.30 | 0.03 | 0.00 | 0.20 | 0.43 |
| | 0.00 | 0.27 | 0.00 | 0.00 | 0.03 | 0.20 |
| | 0.00 | 0.10 | 0.00 | 0.00 | 0.03 | 0.13 |
| | 0.00 | 0.27 | 0.03 | 0.00 | 0.27 | 0.37 |
| | 0.00 | 0.13 | 0.00 | 0.00 | 0.07 | 0.20 |
| | 0.00 | 0.07 | 0.03 | 0.00 | 0.07 | 0.07 |
| | 0.00 | 0.13 | 0.00 | 0.00 | 0.07 | 0.17 |
| | 0.46 | 0.57 | 0.57 | 0.60 | 0.63 | 0.83 |
| | 0.50 | 0.40 | 0.76 | 0.30 | 0.60 | 0.53 |
| | 0.33 | 0.23 | 0.43 | 0.10 | 0.27 | 0.43 |
| | 0.35 | 0.17 | 0.30 | 0.07 | 0.13 | 0.33 |
| | 0.00 | 0.00 | 0.00 | 0.00 | 0.07 | 0.17 |
| | 0.00 | 0.03 | 0.00 | 0.10 | 0.23 | 0.13 |
| | 0.06 | 0.27 | 0.37 | 0.10 | 0.50 | 0.37 |
| | 0.00 | 0.00 | 0.00 | 0.03 | 0.10 | 0.07 |
| | 0.00 | 0.03 | 0.03 | 0.03 | 0.20 | 0.20 |
| | 0.00 | 0.00 | 0.00 | 0.00 | 0.03 | 0.13 |

## G  The Necessity of Learning the Finding-skill

We demonstrate the exploration difficulty of learning skills in Minecraft. Figure 7 shows that a random policy can only travel to a distance of 5 blocks on plains within 500 steps. Since trees are rare on the plains and usually have > 20 distances to the player, an RL agent starting from a random policy can fail to collect logs on plains.

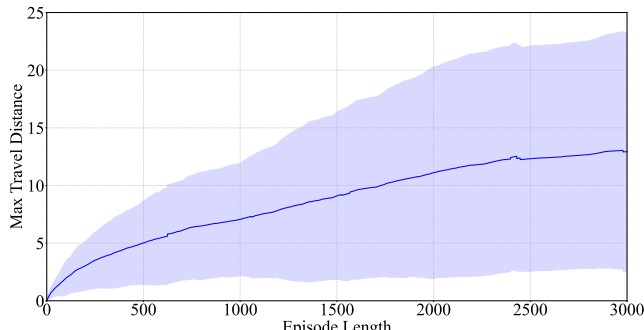

Figure 7: Maximal travel distance to the spawning point a random policy could reach in Minecraft, under different episode lengths. We test for 100 episodes, with different randomly generated worlds and agent parameters. Note that all Manipulation-skills we trained have episode lengths less than 1000 to ensure sample efficiency.

In Table 9, we compare the travel distances of a random policy, a hand-coded walking policy, and our Finding-skill pre-trained with RL. We find that the Finding-skill has a stronger exploration ability than the other two policies.

Table 9: Maximal travel distance on plains of a random policy, a hand-coded policy which always takes forward+jump and randomly turns left or right, and our Finding-skill.

| Episode length | 200 | 500 | 1000 |
|---|---|---|---|
| Random Policy | $3.0 \pm 2.1$ | $5.0 \pm 3.6$ | $7.1 \pm 4.9$ |
| Hand-coded Policy | $7.1 \pm 2.7$ | $11.7 \pm 4.4$ | $18.0 \pm 6.6$ |
| Finding-skill | $12.6 \pm 5.6$ | $18.5 \pm 9.3$ | $25.7 \pm 12.1$ |

## H   Training Manipulation-skills without Nearby Items

For all the Manipulation-skills that are trained with specified environments in the paper, we use the Go-Explore-like approach to re-train them in the environments without target items initialized nearby. In a training episode, the pre-trained Finding-skill explores the environment and finds the target item, then the policy collects data for RL training. In the following, we denote the previous method as Plan4MC and the new method as Plan4MC-go-explore.

Table 10 shows the maximal success rates of these skills over 100 training epochs. We find that all the skills trained with Go-Explore do not fail and the success rates are comparable to the previous skills. This is because the Finding-skill provides good environmental initialization for the training policies. In Milk and Wool, Plan4MC-go-explore even outperforms Plan4MC, because the agent can be closer to the target mobs in Plan4MC-go-explore.

Table 11 shows the test performance of Plan4MC on the four task sets. We find that Plan4MC-go-explore even outperforms Plan4MC on three task sets. This demonstrates that the skills trained with Go-Explore can generalize well to unseen environments.

Table 10: Training success rates of the Manipulation-skills under the two environment settings. Results are the maximal success rates averaged on 100 training epochs.

| Skill | | | | | | |
|---|---|---|---|---|---|---|
| Plan4MC | 0.50 | 0.27 | 0.21 | 0.30 | 0.56 | 0.47 |
| Plan4MC-go-explore | 0.82 | 0.34 | 0.22 | 0.19 | 0.25 | 0.71 |

Table 11: Average success rates on the four task sets of Plan4MC, with the Manipulation-skills trained in the two settings.

| Task Set | Cut-Trees | Mine-Stones | Mine-Ores | Interact-Mobs |
|---|---|---|---|---|
| Plan4MC | 0.417 | 0.293 | 0.267 | 0.320 |
| Plan4MC-go-explore | 0.543 | 0.349 | 0.197 | 0.383 |

We further study the generalization capabilities of learned skills. Table 12 shows the test success rates of these skills in the 40 tasks and the generalization gap. We observe that Plan4MC-go-explore has a small generalization gap in the first four mob-related skills. This is because Plan4MC-go-explore uses the same policy for approaching the target mob in training and test, yielding closer initial distributions for Manipulation-skills. We find that in Harvest Log, Plan4MC-go-explore often finds trees that have been cut before. Thus, it is more difficult to harvest logs in training, and the test success rate exceeds the training success rate.

Table 12: The test success rates of the skills in solving the 40 tasks, and the generalization gap (test success rate - training success rate).

| Skill | | | | | | |
|---|---|---|---|---|---|---|
| Plan4MC | 0.71(+0.21) | 0.26(-0.01) | 0.27(+0.06) | 0.16(-0.14) | 0.33(-0.23) | 0.26(-0.21) |
| Plan4MC-go-explore | 0.86(-0.04) | 0.47(+0.13) | 0.16(-0.06) | 0.16(-0.03) | 0.45(+0.20) | 0.47(-0.24) |

# I  Discussion on the Generalization of Plan4MC

Plan4MC contributes a pipeline combining LLM-assisted planning and RL for skill acquisition. It is widely applicable in many open-world domains (Brohan et al., 2023; Li et al., 2023), where the agent can combine basic skills to solve diverse long-horizon tasks.

Our key insight is that we can divide a skill into fine-grained basic skills, thus enabling acquiring skills sample-efficiently with demonstration-free RL. The Finding-skill in Plan4MC mitigates the exploration issue of RL in vast open-world environments, enabling efficient learning of subsequent skills within a small region. It can be replaced with any learning-to-explore RL policy, or a navigation system in robotics. As an example, for embodied AI tasks, a robotic skill can be defined with action (pick/drop/open) + object. Such a skill involves navigation, arm positioning, and object manipulation, which is challenging to learn with RL in vast scenes. If we develop a navigation policy as the Finding-skill to explore for skill-related objects, the subsequent arm positioning and manipulation skills can be acquired with demonstration-free RL more efficiently. This is because better initial states are provided where objects are close to the agent and the exploration difficulty is substantially reduced.

Our experiments on learning skills in Minecraft demonstrate that object-based intrinsic rewards improve sample efficiency. Figure 8 shows that both MineCLIP reward and distance reward have a positive impact on skill reinforcement learning. This gives motivation to use vision-language models, object detectors, or distance estimation for reward design in skill learning.

For planning, our approach is a novel extension of LLM-based planners, which incorporates LLM knowledge into a graph-based planner, improving planning accuracy. It can be extended to settings where the agent's state can be abstracted by text or entities.

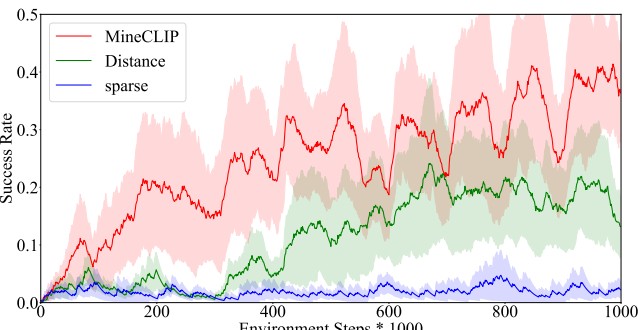

Figure 8: Using different intrinsic rewards for training Harvest Milk with PPO. Results are averaged on 3 random seeds.

## J    Comparison to Other Agents in Minecraft

Table 13 presents a systematic comparison of Plan4MC with other existing agents in Minecraft. Notably, Plan4MC distinguishes itself as the first RL agent to successfully master a wide range of tasks in Minecraft, requiring minimal human supervision and achieving this with a feasible number of environment steps.

Table 13: A systematic comparison between Plan4MC and existing agents in Minecraft.

| Method | Human Prior | Learning Paradigm | Environment Steps | Multi-task |
|---|---|---|---|---|
| HRL methods (Kanervisto et al., 2022) | Expert data | Imitation + RL | 8M | No |
| Dreamer-v3 (Hafner et al., 2023) | -- | RL | 100M | No |
| VPT (Baker et al., 2022) | Expert data | Imitation + RL | 16B | No |
| Voyager (Wang et al., 2023a) | Expert policy | Hard-coded skills | -- | Yes |
| Plan4MC | Reward | RL | 7M | Yes |

