# OpenReview forum: "Skill Reinforcement Learning and Planning for Open-World Long-Horizon Tasks"
_TMLR — Rejected by TMLR_

### Review · Reviewer_M6D2 · 2024-03-03

**Summary Of Contributions:**

The authors propose PlanMC, a method for learning to solve an open-ended set of tasks by learning base skills and then using LLMs to generate graph-based relationships between these skills to figure out how to use them to solve arbitrary new tasks.

**Audience:**

Yes

**Claims And Evidence:**

Yes

**Requested Changes:**

I would like the main issues in the weaknesses section addressed before considering this for acceptance, most importantly:

- The lack of another LLM-guided skill learning baseline
- Lack of any standard deviations
- clarity issues
- better motivating the paper to justify the very specific-to-minecraft method and pre-dfined skills + reward functions vs pre-defined skills + demonstrations.

**Strengths And Weaknesses:**

### Strengths:

**Results:** The results on the Minecraft tasks seem pretty convincing over the listed baselines

**Relevance:** This paper is probably very relevant to the TMLR community and presents research in a topic that is likely to have significant impact on the field.

**Motivation/Framing:** The paper is well-motivated in that it aims to solve long-horizon tasks by learning a large set of tasks in an open-ended manner with LLM guidance.

### **Weaknesses:**

**Clarity:**

- in the intro about LLMs: “fixing their uncontrollable mistakes requires careful prompt engineering” I don’t think these mistakes are uncontrollable if they are fixed (i.e., controlled) via prompt engineering. Uncontrollable failures or mistakes is mentioned a couple times throughout the paper, you should define this term
- Train/test split information should be in the main paper in sec 5.2.
- 4.2 isn’t explained that well, it would be more clear if a simplified version of the pseudocode showing how the algorithm iterates between skill planning and execution is put at the end of 4.2.
- Perhaps the authors could add a method summary paragraph at the end of Section 4 to tie everything together better.

**Method:**

- As it stands, the method is tailored to Minecraft and has not been demonstrated to work on any other environment. The formulation of basic skills of “Finding,” “Manipulation,” and “Crafting” is minecraft-specific, along with using the pre-defined grid state-count to train the high-level FIND policy (this would likely not be applicable to household navigation, for example, due to walls, rooms, and object clutter). There are also many Minecraft-specific reward function changes for Minecraft specific skills in Appendix C.

**Claim/Framing:**

- The authors claim to not require use of demonstrations, but they also hand-define the base skills to learn and then tune the reward functions to make it work. I’m not sure which one is actually more difficult/hard to scale, as both are time and effort-intensive. The introduction can be motivated better by stating *why* they would want to avoid demonstrations here.

**Experiments/Analysis:**

- How dependent is this algorithm on LLM performance? what would it look like if an open-source LLM was used instead of GPT-3.5? I’d imagine that since GPT is very large, it’s able to memorize very specific minecraft information that allows it to produce skill graph information that smaller LLMs cannot.
- There are no standard deviations listed despite multiple random seeds.
- There are actual joint skill learning + LLM baselines to compare to, but any such comparisons are missing from this paper. I think the most relevant is DECKARD: [DECKARD Agent](https://deckardagent.github.io/).
    - This does use demonstrations to *pre-train* the policy, but it’s still good to have an idea of an LLM-guided skill learning baseline with the same environment setup/assumptions as the introduced method.
- There should be an example prompt listed for the LLM planner baseline.

**Minor Issues:**

- “environmental steps” → environment steps
- Some missing related works:
    - LiFT: [LiFT: Unsupervised Reinforcement Learning with Foundation Models as Teachers (gimme1dollar.github.io)](https://gimme1dollar.github.io/lift/), uses LLMs and VLMs to perform reward-free open-ended skill learning
    - BOSS: [Bootstrap Your Own Skills: Learning to Solve New Tasks with LLM Guidance (clvrai.github.io)](https://clvrai.github.io/boss/), uses LLMs to guide skill chaining for open-ended skill learning

**Questions:**

- “Note that Plan4MC totally takes 7M environmental steps in training..” does 7M include the pre-training time for everything? Like the DQN low-level policy and then the high-level goal reaching policy for Finding.
    - I think this comparison shouldn’t be highlighted so much because there are different assumptions in this method from, for example, Hafner et al. 2023. This method specifically has a lot of handcrafted modifications for Minecraft that Hafner et al. 2023 doesn’t.

---

> ### Author Response · Authors · 2024-04-09
>
> ### Thank you for your valuable feedback. We will first respond to the requested changes and then address other issues. We welcome further discussion and additional questions.
>
> ### Main issues in the requested changes:
>
> **Q1:** The lack of another LLM-guided skill learning baseline.
>
> **A1:** DECKARD employs a similar LLM-based approach to generate relations between skills. The differences between our work and DECKARD include the use of GPT-3.5 —- a more advanced LLM, different prompt design, and particularly, our design of fine-grained basic skills augmented by the Finding-skill. This last point marks the most significant difference. Our DECKARD-equivalent configuration, 'Plan4MC w/o Find-skill', has been analyzed within our ablation study.
>
> Also, our baseline named 'LLM Planner' in the main results directly compares our LLM-assisted planning approach with recent works that directly utilize LLMs for planning.
>
> **Q2:** Lack of any standard deviations.
>
> **A2:** In practice, for deploying and testing our agent, we can select the best RL policy for each skill across various training episodes and seeds, which is why our results focus on average test results across tasks and test episodes, rather than including standard deviations over different RL seeds. For each skill, we train with a single seed and select the checkpoint with the highest success rates on smoothed training curves, as detailed in Section C of the Appendix.
> This method aligns with practices in the field, where, due to the low simulation speed of the Minecraft game, other significant works such as VPT [1] also report outcomes based on a single RL training seed.
>
> **Q3:** Clarity issues.
>
> **A3:** We appreciate your comments to improve the clarity of our paper. Here, we respond to each point and revise the corresponding parts in the paper.
>
> - **"Uncontrollable mistakes" for LLM planners:** Upon reflection, we acknowledge that the term 'uncontrollable' may not accurately convey the essence of the challenges posed by LLM planners. The unpredictability of mistakes —- given a specific prompt, it's challenging to foresee the errors an LLM planner might generate —- is the core issue at hand.  Designing a successful LLM planner involves iterative cycles of running the agent in the environment, identifying errors, and meticulously adjusting the prompt to mitigate these errors. This process is notably time-consuming.
> To more accurately describe this, we have updated the term to 'unpredictable mistakes' in the paper and incorporated an explanation.
>
> - **Train/test split:** In our settings, we do not employ a train/test split. All tasks in Section 5.2 are long-horizon tasks for test. Once the skill graph has been generated and the RL skills have been trained, the agent is equipped to address all mentioned tasks directly.
>
> - **Pseudo code showing the iteration between planning and execution; Add a method summary paragraph:** Thanks for your suggestion! We have presented the pseudocode at the end of Section 4.2 and added a method summary at the end of Section 4.
>
>
> **Q4:** Better motivating the paper to justify the very specific-to-Minecraft method. The method has not been demonstrated to work on any other environment.
>
> **A4:** While our implementation involves Minecraft-specific designs, our core contribution lies in an agent framework for general open-world tasks. This framework is anchored by two key technical innovations: (1) The introduction of a general Finding-skill to accelerate learning all skills with RL. (2) The use of an LLM for skill graph generation before the planning stage.
>
> Minecraft-specific techniques, such as MineCLIP rewards and hierarchical RL, are supportive rather than central to our contribution. For example, to adapt our framework to robotic tasks, the hierarcahical RL model for the Finding-skill can be replaced with a navigation system.
>
> To illustrate the potential applicability of our framework to other domains, we have included a detailed example in Section I of the Appendix. We described how a robotic agent could leverage an LLM for skill planning, employ a navigation policy to locate skill-related objects in the vast world, and utilize RL policies for learning contact-rich manipulation tasks.

---

> > ### Author Response · Authors · 2024-04-09
> >
> > **Q5:** RL + reward functions vs demonstrations.
> >
> > **A5:** While our approach uses dense rewards to enable demonstration-free RL, the design of these rewards is simple and general, making it potentially extendable to other domains. Given our basic assumption that each skill is associated with a specific item in the environment, our reward functions are constructed with an item-centric focus. For example, the distance reward is computed based on the distance to the target item, and the depth reward measures the distance to underground ores. Such reward functions are readily applicable in other open-world games or robotic tasks, with access to item states provided within the simulator.
> >
> > Conversely, obtaining demonstrations can often be challenging. To train Minecraft agents using demonstrations, VPT [1] requires gameplay recordings from hundreds of human players. In other domains, such as robotic control, expert demonstrations may be even more difficult to collect.
> >
> > Hence, we advocate that RL based on dense reward functions offers a more universally applicable solution than relying on expert demonstrations.
> >
> > ### Other issues:
> >
> > **Q6:** How dependent is this algorithm on LLM performance?
> >
> > **A6:** The correctness of the skill graph depends on the performance of the LLM. While GPT-3.5 exhibits high accuracy in our work, a similar approach in DECKARD using OpenAI's Codex model achieves a lower accuracy rate of 67%. We have discussed potential approaches to fix the LLM's mistakes in Section 8.
> >
> > Importantly, once the skill graph has been verified and any inaccuracies corrected, our algorithm's planning phase operates independently of the LLM. This ensures that subsequent steps are insulated from the potential impact of LLM mistakes.
> >
> > **Q7:** There should be an example prompt listed for the LLM planner baseline.
> >
> > **A7:** We acknowledge the importance of providing the prompt for the baseline. The prompt will be included in Section D of the Appendix.
> >
> > **Q8:** Minor issues.
> >
> > **A8:** Thanks for pointing these out! We have corrected the typo and properly cited all related works.
> >
> > **Q9:** Does 7M steps include the pre-training time for everything? The comparison of sampling steps to Dreamer-v3 should not be highlighted.
> >
> > **A9:** Yes, the total of 7M environment steps encompasses the entire training process, including the training of the Finding-skill and all other skills.
> >
> > We agree that directly comparing our approach with Dreamer-v3 is not fair due to differing assumptions. However, we believe it remains important to emphasize the sample efficiency of our agent, particularly because this level of efficiency facilitates practical experimentation on lab machines, typically at simulation frame rates under 30 fps.
> >
> > [1]  Video pretraining (vpt): Learning to act by watching unlabeled online videos, 2022.

---

> > > ### Comment · Reviewer_M6D2 · 2024-04-16
> > >
> > > Thanks for the response and addressing many of my concerns. Some followup responses below:
> > >
> > > > Our DECKARD-equivalent configuration, 'Plan4MC w/o Find-skill', has been analyzed within our ablation study.
> > >
> > > I'm not convinced on the baselines, because DECKARD != "Plan4MC w/o Find-Skill," as that is an ablation that doesn't include specific proposals that DECKARD made for their method (if I recall correctly, the way skills are explored and learned is quite different). There are more significant differences than just the use of "Find-Skill" or swapping the LLM.
> > >
> > > > In practice, for deploying and testing our agent, we can select the best RL policy for each skill across various training episodes and seeds, which is why our results focus on average test results across tasks and test episodes, rather than including standard deviations over different RL seeds. For each skill, we train with a single seed and select the checkpoint with the highest success rates on smoothed training curves, as detailed in Section C of the Appendix. This method aligns with practices in the field, where, due to the low simulation speed of the Minecraft game, other significant works such as VPT [1] also report outcomes based on a single RL training seed.
> > >
> > > This is fine, thanks for referencing this. Can this be pointed out directly in the experiments section of the paper to make it very clear?

---

> > > > ### Author Response · Authors · 2024-04-18
> > > >
> > > > Thank you for your detailed follow-up.
> > > >
> > > > Regarding DECKARD: We have thoroughly reviewed both the paper and the code of DECKARD. Its skill learning approach also involves training RL in environments where prerequisite items are provided. DECKARD utilizes VPT trained from human gameplay data to enhance policy exploration, while our approach learns without such demonstrations. For the planning approach, there are only slight differences in the LLM, the prompt design, and the details within the search algorithm. Therefore, 'Plan4MC w/o Find-Skill' can be regarded as a rough equivalent to the baseline using DECKARD’s planning approach + our pre-trained skills. To address your concerns, we are using DECKARD’s planning code to implement this baseline more accurately and will present the results in our final revision.
> > > >
> > > > Regarding seeds in experiments: We have added a detailed explanation in Section 5.2.
> > > >
> > > > We thank you again for your suggestions!

---

### Review · Reviewer_ZW4C · 2024-03-16

**Summary Of Contributions:**

This paper proposes Plan4MC, a hierarchical framework that decomposes solving Minecraft tasks into learning basic skills with intrinsic rewards and planning over the skills with a skill graph and a skill search algorithm. Plan4MC outperforms baselines and ablations on 40 simple Minecraft tasks, but fall shy of state-of-the-art models like VPT.

**Audience:**

Yes

**Claims And Evidence:**

No

**Requested Changes:**

1. Finding-skill (navigation and search) does not seem novel as claimed in the paper. There have been many algorithmic solutions to "finding" items. The authors are suggested to modify the paper to downweigh the claim on "novelty" of finding items.
2. The authors are suggested to include a discussion on how failures/errors in building the graphs could be handled.
3. The authors are suggested to include discussion on the results falling shy of SOTA in Minecraft

**Strengths And Weaknesses:**

1. The paper is clear to follow, and the idea of integrating RL and LLM planning may be an interesting direction for the community.
2. Hierarchical RL implementations may be important for solving challenging problems like Minecraft.

Weaknesses:
1. Results appear too contrived. Cut-Trees, Mine-Stones, Mine-Ores, Interact-Mobs can be completed by human players within 20 minutes of gameplay.
2. It seems that after all decomposing/planning, the agent performs a lot shy of the SOTA, VPT model.
3. The planner is just a DFS on a small set of primitives (referred to as skills) generated by the LLM. It does not seem hard for a human to hand-code such a set of primitives from the ground-truth game specifications.
4. Section 3.1, line 114~115, And Appendix G. Claim: finding things like trees is very hard in Minecraft. This appears to be an overstatement. Authors demonstrated the "distance traveled" of the proposed hierarchical policy compared to a random policy and a "go-forward" hand-coded policy. It appears the hard-coded go-forward policy is only 30% less efficient compared to the complicated hierarchical policy (Table 9).

---

> ### Author Response · Authors · 2024-04-09
>
> ### Thank you for your valuable feedback. We will first respond to the requested changes and then address other issues. We welcome further discussion and additional questions.
>
> ### Requested Changes:
>
> **Q1:** Downweigh the claim on "novelty" of finding items. It appears the hard-coded go-forward policy is only 30% less efficient compared to the complicated hierarchical policy.
>
> **A1:** We acknowledge that the hierarchical RL approach used to implement the Finding-skill in Minecraft may not, in itself, represent a novel contribution. However, the introduction of the Finding-skill as part of our framework for reducing sample complexity in open-world RL is a distinctive contribution of our work. The main finding and motivation of our work, as described in Section 2.2, shows the inefficiency of RL in the vast open-world environment. The introduction of the Finding-skill and fine-grained skill decomposition is significant to reduce the sample complexity in skill learning.
>
> Regarding the comparison with the hand-coded policy shown in Table 9, it's important to note that this policy incorporates human intuition about navigation in Minecraft. Nevertheless, our hierarchical RL-based Finding-skill yields a 1.5x increase in travel distance compared with the hand-coded policy, corresponding to approximately a 2.3x greater area explored and significantly enhancing the likelihood of successfully finding the target item.
> We emphasize that hierarchical RL, while being our optimal implementation of the Finding-skill in Minecraft, is not presented as the primary contribution of our work. Our framework is designed to be adaptable, allowing for different implementations of the Finding-skill in different domains, as elaborated in Section I in the Appendix.
>
>
> **Q2:**  Include a discussion on how failures/errors in building the graphs could be handled.
>
> **A2:** We appreciate your suggestion to include such a discussion. We have incorporated this in Section 8.
>
> **Q3:** Results compared with SOTA in Minecraft. ("The agent falls shy of VPT.")
>
> **A3:** It is important to note that our agent and VPT are designed for fundamentally different settings. VPT relies on a substantial volume of labeled expert data (approximately 3000 hours of early gameplay videos) for finetuning, whereas our method is focused on RL in open-world environments without the aid of demonstrations.  Additionally, VPT requires a considerable number of environment steps —- 16 billion for Diamond and 8 billion for Iron-pickaxe. Such requirements pose significant challenges for implementation on a lab machine, which typically runs game simulations at about 30 fps.
> In contrast, our agent effectively solves all Tech-Tree tasks leading up to the Iron-pickaxe milestone within 7 million environment steps.
>
> We have included a systematic comparison between our agent and existing agents in Minecraft in Section J of the Appendix. This comparison illustrates the distinctive challenges and settings of our work, underscoring the efficiency and novelty of our approach.
>
> ### Other issues:
>
> **Q4:** Results appear too contrived. Cut-Trees, Mine-Stones, Mine-Ores, Interact-Mobs can be completed by human players within 20 minutes of gameplay.
>
> **A4:** We respectfully disagree with this critique. A few minutes of gameplay corresponds to over 10K actions and over 10 subtasks. This complexity instead underlines the significant challenge these tasks pose to an RL agent.  Furthermore, similar tasks have been utilized by concurrent research [1,2] to benchmark their agents, demonstrating the relevance and difficulty of these tasks in the field. Agents in earlier MineRL competitions have faced considerable difficulties in obtaining an Iron-pickaxe, even with access to expert demonstrations.
>
> **Q5:** It does not seem hard for a human to hand-code the skill graph from the ground-truth game specifications.
>
> **A5:**  Indeed, the creation of a skill graph based on ground-truth game specifications can be straightforward with human knowledge of the game's high-level skills and items. However, our approach leverages LLMs to automate the generation of skill graphs, thereby minimizing the need for manual input and potentially introducing scalability benefits.
>
> By utilizing LLM-based planners, we shift the primary challenge in developing agents for open-world environments to mastering low-level control in different skills. This highlights a main contribution of our work that introduces the Finding-skill to accelerate RL.
>
> [1] Describe, explain, plan and select: Interactive planning with large language models enables open-world multi-task agents, 2023
>
> [2] Do embodied agents dream of pixelated sheep?: Embodied decision making using language guided world modelling, 2023.

---

### Review · Reviewer_Gax7 · 2024-03-22

**Summary Of Contributions:**

This paper proposes skill learning and planning for Minecraft long-horizon tasks. It first designs three basic skills manually and a hierarchical skill learning method is used for Finding-skill, and PPO with self-imitation learning is used for the other two skills learning. The planning uses an LLM to construct the skill graph first and the Depth-Search-First approach.

**Audience:**

Yes

**Claims And Evidence:**

Yes

**Requested Changes:**

Please refer to the pros and cons.

Other requested changes,

It is suggested to include an explicit assumption (what assumptions are necessary) and a limitation discussion.

**Strengths And Weaknesses:**

Strengths:
1. The focus of this paper is worth investigating as long-horizon tasks are more realistic and still challenging for RL.
2. Results on long-horizon tasks are promising.

Weaknesses:
1. The novelty of this paper is moderate. This paper integrates a lot of existing techniques, hierarchical RL, MineClip, DSF, etc, making it hard to assess the contributions
2. Some technical details are unclear. For example, the authors mentioned using RL to learn the Finding-skill is difficult in the Introduction section. However, the solution is still a hierarchical RL approach with the high-level PPO and low-level pre-trained fixed DQN. In Section 3.1, it said 'until a target item is detected in its lidar observations'. It is unclear how the target item is defined and how to achieve this without this information (lidar observations).
3. It is still not an effective exploration approach from the reviewer's perspective. 1) The authors assume the target items are uniformly distributed, which is not a realistic assumption. 2) Dividing the surface into discrete grids is not feasible for a very large state space. Perhaps some hierarchical navigation approaches could provide some insights [1-3].
4. The limitation of this approach is not discussed. Given a lot of pre-defined things (skill definition, skill graph, lidar observations), how does the proposed approach generally apply to other domains?

[1] Exploration in deep reinforcement learning: a comprehensive survey

[2] Active Hierarchical Exploration with Stable Subgoal Representation Learning

[3] Generating Adjacency-Constrained Subgoals in Hierarchical Reinforcement Learning

---

> ### Author Response · Authors · 2024-04-09
>
> ### Thank you for your valuable feedback. We have carefully considered your comments and have addressed each point below. We welcome further discussion and additional questions.
>
> **Q1:** This paper integrates a lot of existing techniques, making it hard to assess the contributions.
>
> **A1:** The primary contribution of our paper is a novel framework that integrates RL and LLM-based planning for learning tasks in complex open-world environments. This framework is distinguished by two key technical innovations: (1) The introduction of a general Finding-skill that provides effective initializations for other Manipulation-skills, significantly accelerating RL in complex open-world settings. (2) The use of an LLM for skill graph generation before the planning stage, which enhances the planner's success rate compared with direct LLM-based planning.
>
> While our implementation within the Minecraft benchmark utilizes techniques like MineCLIP rewards and hierarchical RL, these elements are supportive rather than central to our contribution. For example, to implement our framework in robotic tasks, the hierarcahical RL model for the Finding-skill can be replaced with a navigation system.
>
> **Q2:** Using RL to learn the Finding-skill is difficult in the Introduction section. However, the solution is still a hierarchical RL approach.
>
> **A2:** In the Introduction, we discuss the challenge of using RL to learn skills in vast open-world environments, where a policy struggles to navigate effectively and find necessary items.  This issue motivates us to develop a special Finding-skill, designed explicitly to address exploration difficulties by enabling the efficient traversal of large areas to find required items. Once the Finding-skill has been acquired, it significantly enhances the initialization for subsequent skills, making it more feasible to learn these skills using RL.
>
> Hierarchical RL for this Finding-skill is our specific implementation within the Minecraft environment. The RL algorithms used for learning other skills are independent of this approach.
>
> We have revised this part in Introduction for better clarity.
>
> **Q3:**  In Section 3.1, it is unclear how the target item is defined and how to achieve this without this information (lidar observations).
>
> **A3:** During test, the goal of the Finding-skill is to find the specific item required by the subsequent skill in the plan. This information is provided in the structured skill information described in Section 4.1.
>
> While Minecraft offers lidar information for detecting such items, our approach can also accommodate environments that do not provide lidar information. In these cases, success for the Finding-skill can be determined through visual observations using computer vision methods, such as vision-language models. This approach is supported by existing literature [1] and is complementary to our work.
>
> **Q4:** It is suggested to include an explicit assumption and a limitation discussion. The Finding-skill is still not an effective exploration approach.
>
> **A4:** We appreciate your insightful suggestions and agree that making our paper more precise regarding the underlying assumptions and limitations is essential. We clarified the assumptions related to tasks, skills, and the inherent challenges of RL in open-world skill learning within Section 2. We have now explicitly stated the assumptions regarding the information provided in the observations and the LLM’s knowledge of skills.
> We have added a discussion on the limitations in Section 8, including the effectiveness of the exploration approach.
>
> **Q5:** Given a lot of pre-defined things (skill definition, skill graph, lidar observations), how does the proposed approach generally apply to other domains?
>
> **A5:** Pre-defined skills with semantic meanings are prevalent in various realistic open-world tasks, including video games and embodied AI domains [2,3]. The skill graph, which is a component in our planning system, is generated using an LLM. While our current implementation utilizes lidar information, this can be replaced with object detectors from the field of computer vision, as previously mentioned.
>
> To illustrate the potential for adaptability of our framework to other domains, we have detailed an example in Section I of the Appendix. We described how a robotic agent could leverage an LLM to plan its skills, employ a navigation policy to locate skill-related objects in the vast world, and utilize RL policies for learning contact-rich manipulation tasks.
>
> [1] Vision-language models as success detectors, 2023.
>
> [2] Do as i can, not as i say: Grounding language in robotic affordance, 2023.
>
> [3] Bootstrap Your Own Skills: Learning to Solve New Tasks with Large Language Model Guidance, 2023.

---

### Author Response · Authors · 2024-04-09

Dear Reviewers,

Thank you very much for your time and the valuable insights provided during the review. In response to your comments and suggestions, we have uploaded a revision of our manuscript. The changes have been highlighted in blue text for ease of review.

---

### Decision · Action_Editor_hXu1 · 2024-04-25

**Recommendation:** Reject

**Comment:**

Fundamentally, the reviewers feel that the techniques are already well studied/known and the domain limited -- therefore so is the generality. They note substantial manual effort is required to get the system to work and are not convinced by the setting -- including the tasks explored, even in relation to the existing literature for minecraft specifically though a larger (e.g. robotics planning) literature exists as well.  Addressing the concerns around generality will be necessary for a revision.

Note specifically
`M6D2`
> As it stands, the method is tailored to Minecraft ... not be applicable to household navigation ... many Minecraft-specific reward function changes for Minecraft specific skills in Appendix C.
which the authors claim is not a concern because it's not the core contribution of their work but again the
>The authors claim to not require use of demonstrations, but they also hand-define the base skills to learn and then tune the reward functions to make it work.

`ZW4C` and `Gax7` also note
> Results appear too contrived..
> The novelty of this paper is moderate.
>  technical details are unclear.

In response to all of these concerns, the authors fall back to arguments about the paper being a framework, but the strength of a framework is it's broad applicability to other domains and none of the reviewers see how that can be done without the same substantial manual effort expended for Minecraft.  Further, even within this domain the results are not as strong as expected and there's a lack of clarity on what about the framework is a new contribution or demonstration for the field.

Rewriting the work to contextualize the authors insights/contributions to justify the framework and then apply the framework to an additional domain would address these concerns.

**Audience:**

Planning research with LMs and in embodied environments.

**Claims And Evidence:**

The aim of this work is skill learning and planning within Minecraft, specifically for long-horizon tasks. Several basic skills are defined and learned via various techniques. The planning works by decomposing tasks into basic skills and then planning over a skill graph -- the LLM can then be used to generate graph relations between skills for novel tasks.

**Resubmission Of Major Revision:**

The authors may consider submitting a major revision at a later time.